# Enhancing Diversity In Parallel Agents:
# A Maximum State Entropy Exploration Story

**Vincenzo De Paola** [1]   **Riccardo Zamboni** [1]   **Mirco Mutti** [2]   **Marcello Restelli** [1]

## Abstract

Parallel data collection has redefined Reinforcement Learning (RL), unlocking unprecedented efficiency and powering breakthroughs in large-scale real-world applications. In this paradigm, $N$ identical agents operate in $N$ replicas of an environment simulator, accelerating data collection by a factor of $N$. A critical question arises: *Does specializing the policies of the parallel agents hold the key to surpass the $N$ factor acceleration?* In this paper, we introduce a novel learning framework that maximizes the entropy of collected data in a parallel setting. Our approach carefully balances the entropy of individual agents with inter-agent diversity, effectively minimizing redundancies. The latter idea is implemented with a centralized policy gradient method, which shows promise when evaluated empirically against systems of identical agents, as well as synergy with batch RL techniques that can exploit data diversity. Finally, we provide an original concentration analysis that shows faster rates for specialized parallel sampling distributions, which supports our methodology and may be of independent interest.

## 1. Introduction

*Exploration*, or taking sub-optimal decisions to gather information about the task, lies at the heart of Reinforcement Learning (RL, Bertsekas, 2019) from its very foundation. When the task of interest is defined through a reward function, exploration shall be carefully balanced with the *exploitation* of the current information to maximize the rewards collected over time (Mnih et al., 2013). When a task is not known at the time of the interaction, exploration becomes an objective *per se*, in order to foster the collection of maximally informative data.

A common formulation of the latter pure exploration setting is *state entropy maximization* (Hazan et al., 2019), in which the agent aims to maximize the entropy of the data distribution induced by its policy. The state entropy objective has been widely studied in the literature, e.g., to provide data collection strategy for offline RL (Yarats et al., 2022; Park et al., 2023), experimental design (Tarbouriech & Lazaric, 2019), or transition model estimation (Tarbouriech et al., 2020), and as a surrogate loss for policy pre-training in reward-free settings (Mutti & Restelli, 2020). All of the above uphold the status of state entropy maximization as a flexible tool for exploration. To this day, however, an important gap remains in how to deploy state entropy maximization on parallel simulators.

Let us picture an illustrative example to guide our thoughts: A robotic arm control task. We may want the robot to be able to perform a wide range of tasks, such as interacting with a diverse set of objects. A good strategy is to exploit a simulator of the robotic arm to learn good manipulation policies in advance, instead of relying on costly online interactions, in which exploration may be unsafe. Since not all of the future tasks may be known in advance, one could opt for state entropy maximization, in order to collect exploration data once and for all. Collecting those data on complex simulators may be inefficient though, resulting in long simulation times. However, one crucial advantage of sampling in simulation rather than the physical system is that the sampling process does not have to be sequential.

In many modern simulators (NVIDIA Isaac Sim; Genesis, 2024; Todorov et al., 2012), the process of learning is accelerated by instantiating vectorized instances of agents and environments simultaneously. Parallelization was shown to be a game-changing approach that allowed for unprecedented performance in standard RL (Espeholt et al., 2018), and the use of a multitude of identical agents operating on copies of the same environment allowed RL systems to tackle increasingly complex and large-scale problems. While the maximum entropy formulation has been widely adopted in prior works, how to better take advantage of parallel exploration agents is still to be investigated.

[1]AIRLAB, Politecnico di Milano, Milan, Italy [2]Technion, Israel Intitute of Technology, Haifa, Israel. Correspondence to: Vincenzo De Paola <vincenzo.depaola@polimi.it>, Riccardo Zamboni <riccardo.zamboni@polimi.it>.

*Proceedings of the 42nd International Conference on Machine Learning*, Vancouver, Canada. PMLR 267, 2025. Copyright 2025 by the author(s).

This is far from being a trivial problem. Zhong et al. (2024) recently showed that not explicitly caring for redundant behaviors among the agents often results in inefficient use of computational resources, which is why parallelization is employed in the first place. Additionally, as highlighted in Rudin et al. (2022), solving complex problems necessitates intelligent strategies for constructing experience that accelerate algorithmic convergence. This challenge is further compounded in scenarios where access to real-world process data is limited, examples are scarce, or simulators are computationally expensive. Following these considerations, our study is centered around the following questions:

> *How should parallel agents optimize their policies*
> *for efficient exploration?*
> *What is the best way to leverage parallelism in state*
> *entropy maximization?*

In this paper, we generally address this question by extending state entropy maximization to incorporate the role of parallel agents, enabling a deeper investigation into their exploration efficiency. First, we introduce a specific problem formulation that can explicitly leverage parallelism. Rather than considering agents independently, we look at them as a whole, and consider the distribution a single one virtual agent would induce if it was following a uniform mixture of the agents' policies. Additionally, we theoretically characterize how the entropy of the distribution of state visits concentrates in the single-agent and parallel-agent cases and demonstrate the significant impact of parallelization on the rate of entropy stabilization and exploration diversity. Then, we shift our attention to how to solve the problem of parallel state exploration *in practice*, by proposing a policy gradient strategy (Williams, 1992) that explicitly optimizes for the objective at hand. Finally, we use illustrative experiments to show that the proposed algorithm is indeed able to optimize for the state entropy and that the induced behavior enables agents to balance individual exploration with inter-agent diversity. This synergy is particularly relevant when integrated with batch RL techniques (Levine et al., 2020).

**Original Contributions.** Throughout the paper, we make the following contributions:

- We define the problem of State Entropy Maximization in Parallel MDPs (Section 3);
- We provide high-probability concentration bounds describing the rate at which the entropy of empirical distributions converges to the entropy of their stationary distributions. The proofs employ novel techniques that may be of independent interest (Section 4);
- We introduce a policy gradient algorithm to take advantage of parallel agents for maximizing the entropy of the visited states (Section 5);

- We provide numerical experiments to corroborate our findings, both in terms of state entropy maximization and batch RL on the collected data (Section 6).

## 2. Preliminaries

Before diving into the technical contributions of our work, we introduce the notation and the relevant background.

**Notation.** We denote $[N] := \{1, 2, \ldots, N\}$ for a constant $N < \infty$. We denote a set with a calligraphic letter $\mathcal{A}$ and its size as $|\mathcal{A}|$. We denote $\mathcal{A}^T := \times_{t=1}^{T} \mathcal{A}$ the $T$-fold Cartesian product of $\mathcal{A}$. The simplex on $\mathcal{A}$ is denoted as $\Delta_{\mathcal{A}} := \{p \in [0,1]^{|\mathcal{A}|} | \sum_{a \in \mathcal{A}} p(a) = 1\}$ and $\Delta_{\mathcal{A}}^{\mathcal{B}}$ denotes the set of conditional distributions $p : \mathcal{A} \to \Delta_{\mathcal{B}}$. Let $X, X'$ random variables on the set of outcomes $\mathcal{X}$ and corresponding probability measures $p_X, p_{X'}$, we denote the Shannon entropy of $X$ as $\mathcal{H}(X) = -\sum_{x \in \mathcal{X}} p_X(x) \log(p_X(x))$ and the Kullback-Leibler (KL) divergence as $D_{KL}(p_X \| p_{X'}) = \sum_{x \in \mathcal{X}} p_X(x) \log(p_X(x)/p_{X'}(x))$.

**Interaction Model.** As a base model for interaction, we consider a finite-horizon Parallel Markov Decision Process (PMDP, Sucar, 2007) without rewards. A PMDP $\mathbb{M}_p = (\mathbb{M}_i)_{i \in [m]}$ consists of $m \in \mathbb{N}^+$ copies of the same MDP $\mathbb{M}_i = (\mathcal{S}, \mathcal{A}, \mathcal{P}, T, \mu)$, composed of a set $\mathcal{S}$ of states and a set $\mathcal{A}$ of actions, which we let discrete and finite with size $|\mathcal{S}|, |\mathcal{A}|$ respectively. A set of $m$ independent agents interact with each copy $\mathbb{M}_i$, according to the following protocol. At the start of an episode, the initial state is drawn from an initial state distribution $s_0^i \sim \mu$. Then the $i$-th agent takes action $a_0^i$ and the state transitions to $s_1^i \sim \mathcal{P}(\cdot|s_0^i, a_0^i)$ according to the transition kernel $\mathcal{P} : \mathcal{S} \times \mathcal{A} \to \Delta_{\mathcal{S}}$. Those steps are repeated until $s_T^i$ is reached, being $T < \infty$ the horizon of an episode. The $i$-th agent takes actions according to a *policy* $\pi^i \in \Delta_{\mathcal{S}}^{\mathcal{A}}$ such that $\pi^i(a^i|s^i)$ denotes the conditional probability of taking action $a^i$ upon observing $s^i$. Overall, we refer with *parallel policy* the collection of policies $\pi_p = (\pi^i)_{i \in [m]}$.[1] Each interaction episode is collected into a *trajectory* $\tau^i := (s_0^i, a_0^i, \ldots, s_{T-1}^i, a_{T-1}^i, s_T^i) \in \mathcal{K} = \mathcal{S}^T \times \mathcal{A}^T$, where $\mathcal{K}$ is the set of all trajectories of length $T$.

**Trajectory and State Distributions in MDPs.** Here we consider a single MDP $\mathbb{M}_i$ for the purpose of expanding the connection between trajectories and distribution over states, temporarily dropping the index $i$ from all the quantities. A policy $\pi$ interacting with $\mathbb{M}_i$ induces a distribution over the generated trajectories. In particular, a set of $n \in \mathbb{N}^+$ trajectories $\tau = (\tau_j)_{j \in [n]}$ are distributed according to a probability measure $p_\pi \in \Delta_{\mathcal{K}}$ defined as:

$$p_\pi(\tau) = \prod_{j=1}^{n} \mu(s_{0,j}) \prod_{t=0}^{T-1} \pi(a_{t,j} \mid s_{t,j}) P(s_{t+1,j} \mid s_{t,j}, a_{t,j}).$$

---

[1] In general, we will denote the set of valid per-agent policies with $\Pi^i$ and the set of joint policies with $\Pi$.

From the realization of these trajectories, one can extract their empirical distribution $\rho_n \in \Delta_{\mathcal{K}}$ defined as $\rho_n(\tau) = \frac{1}{n} \sum_{j=1}^{n} \mathbb{1}(\tau_j = \tau)$, where $\tau_j$ is the trajectory at episode $j$ in the history of interactions. Additionally, a trajectory obtained from an interaction episode induces an empirical distribution over states $d_n \in \Delta_{\mathcal{S}}$ given by

$$d_n(s) = \frac{1}{nT} \sum_{j=1}^{n} \sum_{t=0}^{T-1} \mathbb{1}(S_{t,j} = s) \in \Delta_{\mathcal{S}}.$$

With a slight overload of notation, we will denote as $d_n \sim p_\pi$ an empirical state distribution obtained from a sequence of trajectories of length $T$, and with $\tau_t \sim p_\pi^t$ a *sub-trajectory* of length $t < T$ drawn from $p_\pi$. Finally, we denote the expectation of the empirical state distribution under the policy $\pi$ as $d_\pi(s) = \mathbb{E}_{d_n \sim p_\pi}[d_n(s)]$, such that $d_\pi \in \Delta_{\mathcal{S}}$ is called the state distribution induced by $\pi$.

**State Entropy Maximization in MDPs.** In standard RL (Sutton & Barto, 2018), an agent interacts with an environment to maximize the (cumulative) reward collected from the MDP. In the absence of a reward, previous works (Hazan et al., 2019; Mutti & Restelli, 2020; Mutti et al., 2021) investigated the effects of optimizing a different metric, namely the state entropy $\mathcal{H}_s$ defined as:

$$\mathcal{H}_s(\pi) \triangleq \mathcal{H}(d_\pi) := -\sum_{s \in \mathcal{S}} d_\pi(s) \log d_\pi(s).$$

More recently, Mutti et al. (2023) noted that only a finite number of episodes (called *trials*) can be drawn in many practical scenarios. When only $n$ trajectories can be obtained, they propose to focus on $d_n$ rather than its expectation $d_\pi$, translating the *infinite-trials* objective of Eq. (2) with the *finite-trials* version

$$\mathcal{J}_s(\pi) \triangleq \mathbb{E}_{d_n \sim p_\pi} \mathcal{H}(d_n).$$

The Jensen's inequality relates the two formulations as $\mathcal{H}_s(\pi) \geq \mathcal{J}_s(\pi)$.

## 3. Problem Formulation

In this section, we extend the concept of state entropy to a novel objective able to incorporate parallel agents interacting with independent copies of the environment. In order to do so, we first need to introduce the following objects:

**Definition 3.1** (Distributions in PMDPs). Let $\mathbb{M}_p$ a PMDP with an interaction over $n \in \mathbb{N}$ episodes, then overall set of trajectories $\tau = \{\tau_j^i\}_{j \in [n]}^{i \in [m]}$ is distributed according to the following probability measure:

$$p_{\pi_p}(\tau) := p_{\pi^1, \dots, \pi^m}(\tau) = \frac{1}{m} \sum_{i=1}^{m} \sum_{j=1}^{n} p_{\pi^i}(\tau_j^i),$$

while the states are distributed according to the following mixture distribution:

$$d_{\pi_p}(s) = \frac{1}{m} \sum_{i=1}^{m} d_{\pi^i}(s).$$

Collecting data from all agents induces an empirical (parallel) distribution over states which we denote as $d_{n,p} \in \Delta_{\mathcal{S}}$, so that it is possible to define the mixture distribution as the expectation of the empirical state distribution under the parallel policies, namely $d_{\pi_p}(s) = \mathbb{E}_{d_{n,p} \sim p_{\pi_p}}[d_{n,p}(s)]$. This object allows us to define objectives specifically designed for PMDPs, that are then able to leverage their structural properties. First of all, we introduce the following:

**Definition 3.2** (Parallel Learning Objective in Infinite Trials). Let $\mathbb{M}_m$ a PMDP, then the corresponding infinite trials objective can be defined as:

$$\mathcal{H}_{s,p}(\pi) \triangleq \mathcal{H}(d_{\pi_p}) := -\sum_{s \in \mathcal{S}} d_{\pi_p}(s) \log d_{\pi_p}(s).$$

Now, while this objective will be shortly shown to enjoy rather interesting concentration properties, we are building over the motivating interest of exploring with a handful of trials. These considerations lead to:

**Definition 3.3** (Parallel Learning Objective in Finite Trials). Let $\mathcal{M}_m$ a PMDP with an interaction over $n \in \mathbb{N}$ episodes, then the corresponding finite trials objective can be defined as:

$$\mathcal{J}_p(\pi_p) \triangleq \mathbb{E}_{d_{n,p} \sim p_{\pi_p}} \mathcal{H}(d_{n,p}).$$

We are now ready to state the novel problem of interest:

**Parallel State Entropy Maximization**

$$\max_{\pi_p \in \Pi} \mathcal{J}_p(\pi_p) = \max_{\pi_p \in \Pi} \left\{ \mathbb{E}_{d_{n,p} \sim p_{\pi_p}} \mathcal{H}(d_{n,p}) \right\} \quad (1)$$

We denote by $\pi_p^\star \in \arg\max_\pi \mathcal{J}_p(\pi)$ a parallel policy that maximizes the parallel states entropy. In the next sections, we will show that Parallel Learning Objectives enjoy extremely nice properties: first of all, they concentrate faster than their non-parallel counterparts, motivating the need for parallel exploration; moreover, they can be easily optimized in a decentralized fashion through policy gradient methods.

## 4. Fast Concentration of the Entropy in Parallel Settings

In this section, we present theoretical results that illustrate how the (infinite-trials) entropy of the state (or trajectory) distribution concentrates. We will use these results to demonstrate the significant impact of parallelization on the rate of entropy stabilization and exploration diversity.

**Key Remarks.** While the following theorems are instantiated for state distributions, they are of broader theoretical interest and apply equally to distributions *over trajectories*.[2] Additionally, we adopt the convention of referring to the entropy and variance of distributions rather than those of random variables, as a slight abuse of terminology, since these quantities are technically defined for random variables but naturally extend to their associated distributions.

We start by analyzing the entropy of state distributions in MDPs, for which the following result holds:

**Theorem 4.1.** *Let $d_\pi$ be the (categorical) distribution induced by $\pi$ over the finite set $\mathcal{S}$ with $|\mathcal{S}| = S$, and let $d_n$ be the empirical distribution obtained from $n$ independent samples drawn from $d_\pi$. Then, for any $\epsilon > 0$, the following bound holds:*

$$\mathbb{P}\left(\mathcal{H}(d_\pi) - \mathcal{H}(d_n) > \epsilon\right) \leq 2S \exp\left(-n\frac{\epsilon^2 \mathrm{Var}(d_\pi)}{2S^3 \mathcal{H}^2(d_\pi)}\right),$$

*where $H(d_n)$ and $H(d_\pi)$ denote the entropy of the empirical and true distributions, respectively, and $\mathrm{Var}(d_\pi) = \sum_{s \in [\mathcal{S}]} d_\pi(s)(1 - d_\pi(s))$ is the variance of a random variable associated with the categorical distribution $d_\pi$. Furthermore, to ensure this concentration with confidence $1 - \delta$, the number of samples $n$ must satisfy the following lower bound:*

$$n \geq \frac{2S^3 \mathcal{H}^2(d_\pi)}{\epsilon^2 \mathrm{Var}(d_\pi)} \cdot \ln\frac{2S}{\delta}.$$

This theorem establishes an upper bound on the probability that the entropy difference between the true and empirical distributions exceeds $\epsilon$. Specifically, the probability of large deviations between these two entropies decreases exponentially with $n$, with the rate of convergence influenced by the entropy of the true distribution $d^\pi$. Notably, as $\lim_{\mathcal{H}(d_\pi) \to 0} \frac{\mathcal{H}^2(d_\pi)}{\mathrm{Var}(d_\pi)} = 0$, distributions with lower entropy require fewer samples for concentration, implying they are easier to approximate empirically. The full proof can be found in Appendix A.

**Primary Insight.** This result suggests a key advantage of parallel exploration: when multiple agents explore the environment simultaneously, each can focus on different regions of the state space. As a result, they induce distributions with lower entropy compared to a single policy covering the entire space.

We now consider the case of PMDPs with $m$ parallel agents, each independently collecting $n/m$ samples from the en-

vironment for a total of $n$ samples. This setting induces mixture distributions $d_{\pi_p}$ and empirical mixture distributions $d_{n,p}$ as defined in the previous sections.

The entropy of the mixture distributions decomposes as follows:

$$\mathcal{H}(d_{\pi_p}) = \frac{1}{m}\sum_{i=0}^{m} \mathcal{H}(d_{\pi^i}) + \frac{1}{m}\sum_{i=0}^{m} D_{KL}(d_{\pi^i} || d_{\pi_p}),$$

where the first term is the average entropy of the individual agent distributions, and the second term captures diversity among these distributions via the KL divergence. By analyzing this decomposition, we can get insights into the potential advantages of parallel exploration versus single-agent exploration. To achieve high-entropy mixture distributions, we have two key strategies:

**Homogeneous Exploration:** All agents follow the same entropy-maximizing policy, leading to a mixture with an entropy equivalent to that of a single agent (the average of the entropies is equal to the single-agent entropy, and the second term in the above decomposition is equal to zero). The primary benefit of parallelism, in this case, is computational speedup rather than sample complexity reduction.

**Diverse Exploration:** Agents follow distinct policies with lower individual entropy, but their induced distributions are maximally different. Here, the average entropy remains low, but the overall mixture entropy is high due to diversity (i.e., high values of the KL divergences).

For example, if $m$ agents induce distributions with disjoint supports, the entropy of the mixture increases by $\log(m)$ while maintaining low individual entropy, enabling faster concentration. However, if the agents' distributions overlap, the average KL divergence term decreases, necessitating higher individual entropies to maintain high mixture entropy, thereby increasing sample complexity.

This analysis shows that parallel exploration is beneficial not only for reducing data collection time but also for improving sample efficiency by enabling broader state-space coverage with fewer samples.

## 5. Policy Gradient for Parallel Exploration with a Handful of Trials

As stated before, a core motivation of this work is addressing the problem of exploration in practical scenarios, where a strong parallelization of the environment is used to overcome the difficulty to sample from complex simulators or even physical instantiations of the agents. In such setting, it

---

[2]The number of samples $n$ will represent the number of episodes for trajectory distributions or the number of steps for state distributions.

is crucial to achieve good performances over a finite number of realizations, and for this reason our attention now shifts towards finite trials objectives. Specifically, we will focus on the single trial case, i.e., $n = 1$, and show that rather than instantiating $m$ copies of the same algorithm, the parallel learning objective allows for an alternative formulation, where agents explicitly leverage parallelization. The core idea lies in the implementation of a policy gradient of the parallel formulation of Eq. 3.3 on a set of parametric policies $\pi_{\theta_i} : \mathcal{S} \to \Delta_{\mathcal{A}}$, where $\theta_i \in \Theta_i \subset \mathbb{R}^d$ is a parameter vector in a $d$-dimensional space, uniquely associated with the agent $i$. The parallel policy is then defined as $\pi_\theta = (\pi_{\theta_i})_{i \in [m]}$.

In the following, we derive the gradient of the parallel objective with respect to the policy parameters. For the sake of simplifying the notation, we will refer to $d_{n,p}$ as $d_p$ for simplicity. Starting from the cost function:

$$\mathcal{J}(\pi_p) = \int p_{\pi_p}(d_p) \mathcal{H}(d_p),$$

we take the derivative with respect to the parameters $\theta_i$:

$$\frac{\partial \mathcal{J}(\pi_p)}{\partial \theta_i} = \frac{\partial}{\partial \theta_i} \int p_{\pi_p}(d_p) \mathcal{H}(d_p).$$

Since $\theta_i$ only appears in $p_{\pi_p}(\cdot)$, we can then write:

$$\frac{\partial \mathcal{J}(\pi_p)}{\partial \theta_i} = \int \frac{\partial p_{\pi_p}(d_p)}{\partial \theta_i} \mathcal{H}(d_p),$$

Now, we can employ standard log-trick arguments after noticing that the parameter $\theta_i$ only appears in agent $i$'s policy, that is conditioned over the respective trajectory $\tau_i$, allowing one to write:

$$\frac{\partial p_{\pi_p}(d_p)}{\partial \theta_i} = p_{\pi_p}(d_p) \sum_{t=0}^{T-1} \frac{\partial \log \pi_{\theta_i}(a_t^i|s_t^i)}{\partial \theta_i},$$

Then, after substituting back into the original integral we get:

$$\frac{\partial \mathcal{J}(\pi_p)}{\partial \theta_i} = \int p_{\pi_p}(d_p) \log \pi_{\theta_i} \mathcal{H}(d_p).$$

defining the so-called *score function* as $\log \pi_{\theta_i} \triangleq \sum_{t=0}^{T-1} \nabla_\theta \log \pi_{\theta_i}(a_t|s_t)$, with $\pi_{\theta_i}(a_t|s_t)$ being the probability for agent $i$ of taking action $a_t$ in state $s_t$ in trajectory $\tau_i$. On the other hand, one should note that the entropy is computed over the empirical distribution $d_p$ induced by *all* the agents. In other words, each agent can independently perform gradient updates via a joint estimation of the entropy feedback. In the following, we estimate the gradient of the parallel objective $\mathcal{J}(\pi_\theta)$ with respect to its policy parameters $\theta_i$ using an unbiased Monte-Carlo estimator over $K \in \mathbb{N}$ sampled parallel trajectories:

$$\nabla_{\theta_i} \mathcal{J}(\pi_\theta) \approx \frac{1}{K} \sum_{k=1}^{K} \left( \sum_{t=0}^{T-1} \nabla_\theta \log \pi_{\theta_i}(a_{t,k}^i|s_{t,k}^i) \right) \mathcal{H}(d_p^k).$$

---

**Algorithm 1**: Policy Gradient for Parallel States Entropy maximization (**PGPSE**)

1: **Input**: Episodes N, Trajectories K, Batch Size B, Learning Rate $\alpha$, Parameters $\theta = (\theta^i)_{i \in [m]}$
2: **for** $e \in \{1, \dots, N\}$ **do**
3:     **for** $itr \in \{1, \dots, B\}$ **do**
4:         **for** $k \in \{1, \dots, K\}$ **do**
5:             $\tau \sim \pi_\theta$    {Sample parallel trajectories}
6:             $\log \pi_{\theta_i} \leftarrow \sum_{t=1}^{T-1} \nabla_\theta \log \pi_\theta(a_t \mid s_t)$
7:             $d_p(s) \leftarrow \frac{1}{km} \sum_{j,i,t=1}^{m,k,T} \mathbf{1}(s_{t,i,j} = s)$
8:             $\nabla_\theta \mathcal{J}(\theta) \mathrel{+}= \log \pi_{\theta_i} \cdot \mathcal{H}(d_p)$
9:         **end for**
10:     **end for**
11:     $\nabla_\theta \mathcal{J}(\theta) \leftarrow \frac{1}{B} \nabla_\theta \mathcal{J}(\theta)$
12:     $\theta \leftarrow \theta + \alpha \nabla_\theta \mathcal{J}(\theta)$
13: **end for**
14: **Output**: Policies $\pi_\theta = (\pi_{\theta^i}^i)_{i \in [m]}$

---

Thus, the partial gradient update consists of an internal summation over the log of the $i$-th policy executed by agent $i$, as it interacts over the horizon $T$, experiencing various action-state combinations. This internal summation is further aggregated across the external summation over $K$ trajectories. This estimator approximates the true gradient by averaging the score function of sampled trajectories, weighted by the entropy of such induced state distribution. By limiting parallel agents to experience only one trajectory per each, we average over a mini-batch $\mathcal{B} = \{(\tau_i)\}_{i=1}^n$ to reduce the variance of the reinforce estimator, with $B = |\mathcal{B}|$ being the batch-size. The pseudo-code of the resulting Algorithm, called *Policy Gradient for Parallel States Entropy maximization* (**PGPSE**), is reported in Algorithm 1.

## 6. Empirical Corroboration

We report numerical experiments in simple yet illustrative domains to demonstrate the advantage of exploration with distinct parallel agents over single agents. The section is organized into three parts:

- **Policy Gradient Optimization**: This part analyzes the results of the learning process based on Algorithm 1, evaluating the performance of the learned policy in terms of normalized entropy and support size.

- **Dataset Entropy Evaluation**: This part takes a deeper look at the entropy of a dataset collected from the policies trained as in the previous section.

- **Offline RL**: This part provides a comparative analysis of how the dataset collected with parallel maximum entropy agents benefits the performance of offline RL.

For all the experiments, we consider variations of discrete grid-world domains, which we depicted in Appendix B (see

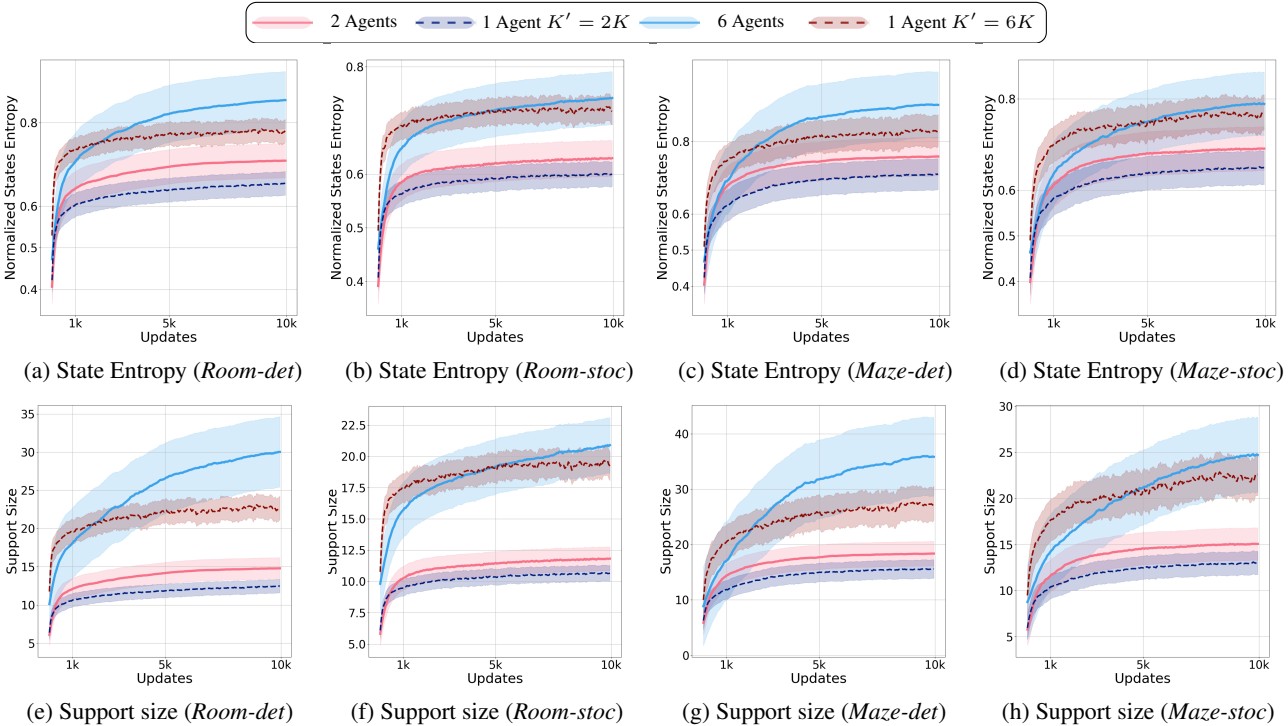

*Figure 1.* The top row (a–d) shows the progression of normalized state entropy across different environments, while the bottom row (e–h) depicts the corresponding size of the support of the entropy. Each plot shows the performance of parallel agents (2 or 6) against single agents taking 2 and 6 trajectories, respectively. We report average results and standard deviation over 5 independent runs.

Figures 5, 6). One family (referred to as *Room*) is composed of two rooms connected by a narrow corridor, with a total of 43 states and 4 actions (one to move in each direction). The other (referred to as *Maze*) is a maze of various connected paths, with same number of total states and actions as before. For each family, we consider a deterministic version (*det*) and a stochastic version (*stoc*) in which the agent's chosen actions are flipped to random actions with some probability. Appendix C reports further experimental details, with additional visualizations and results.

**Policy Gradient Optimization.**     First of all, we test the ability of PGPSE (Algorithm 1) in addressing maximum state entropy exploration with parallel agents. We run multiple instances of the algorithm with an increasing number of agents $m \in \{2, 4, 6\}$.[3] As a baseline, we consider the performance of the same algorithm training a single agent for the finite-trial objective (Eq. 2), with a number of trials $K'$ matching the number of trajectories taken by the parallel agents. We compare the results in terms of (normalized) state entropy and the size of the entropy support (i.e., the number of unique visited states) in all the domains.

Figure 1 reports the resulting learning curves, obtained over five independent runs. For a fair comparison of al-

gorithms with varying number of agents, the performance is reported as a function of the policy updates, which translate to $B \times m \times K \times T$ total interactions with the environment. Across the board, the performance of parallel agents (solid lines) consistently outperforms the single agent counterparts (dashed lines). Indeed, they achieve higher entropy values, thanks to their ability to explore more states as it is evident from the larger size of the support. This behavior can be attributed to the intrinsic diversity introduced by parallel agents: To optimize the team state distribution, each agent learns an individual behavior that is synergic with the others.

**Dataset Entropy Evaluation.**     Then, we tested the capabilities of the trained policies in collecting diverse experiences, by analyzing the properties of a single dataset. Figure 2 illustrates the normalized empirical state entropy of each dataset collected by the $m$ agents, compared against the entropy of the dataset collected by a single agent over $K = m$ trajectories (trained to maximize the entropy on the corresponding number of trajectories), and a random policy.

To construct these datasets, we sample one trajectory per agent from the parallel policies, evaluating them across five seeds. This procedure reveals that datasets generated through parallel exploration exhibit higher entropy than those collected by a single agent or a random policy. It is noteworthy that experience collected in parallel also ex-

---

[3]In the main text we omit the result with 4 agents, which are reported in Figure 7 of the Appendix.

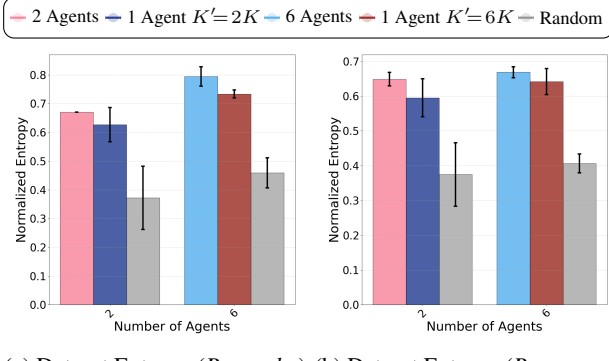

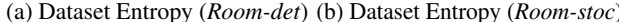

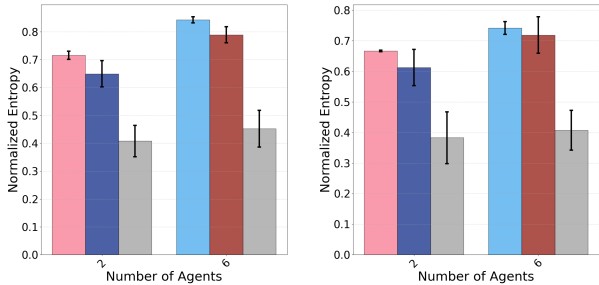

(a) Dataset Entropy (*Room-det*) (b) Dataset Entropy (*Room-stoc*)

(c) Dataset Entropy (*Maze-det*) (d) Dataset Entropy (*Maze-stoc*)

*Figure 2.* Entropy evaluation of the dataset generated by sampling from parallel policies, compared against both random and single-baselines across different environments. With the notation $K' = mK$ we denoted the set in which the single agent had access to a multiple $m$ of trajectories available to the parallel agents. We report average and standard deviation over 5 independent runs.

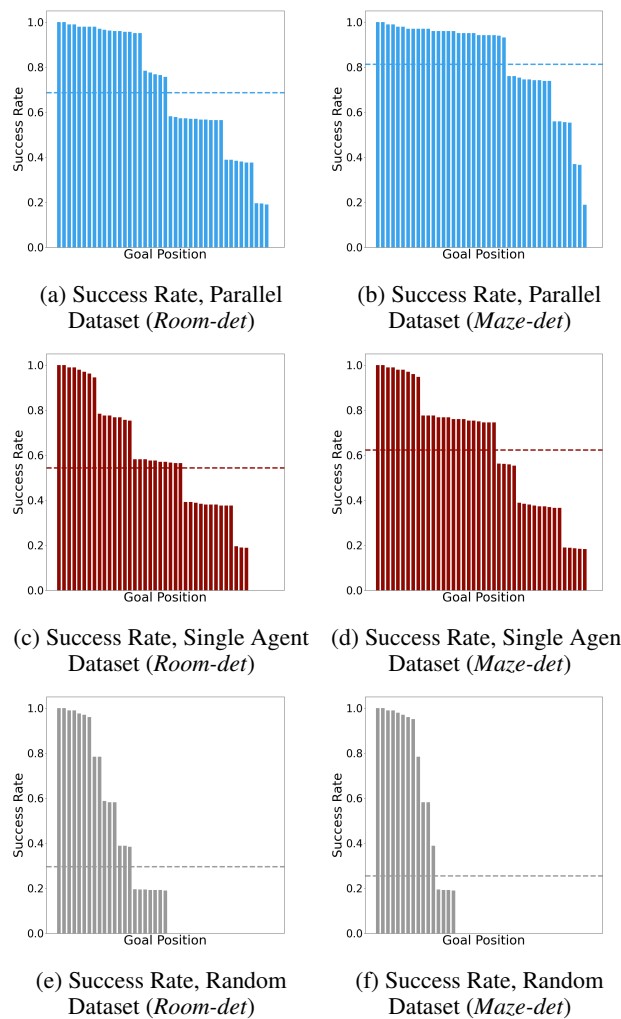

(a) Success Rate, Parallel Dataset (*Room-det*)   (b) Success Rate, Parallel Dataset (*Maze-det*)

(c) Success Rate, Single Agent Dataset (*Room-det*)   (d) Success Rate, Single Agent Dataset (*Maze-det*)

(e) Success Rate, Random Dataset (*Room-det*)   (f) Success Rate, Random Dataset (*Maze-det*)

*Figure 3.* Offline Q-learning success rate for various goal positions: Each subplot depicts the agent's ability to reach a goal state positioned at a different location (one for each state). The results refer to datasets obtained over 5 independent runs.

hibits minimal variance across runs, indicating that agents tend to generate trajectories in a quasi-deterministic manner, preserving consistency in the collected data.

**Offline RL.** Finally, we address the following question: *Can datasets collected with parallel maximum entropy agents benefit offline RL algorithms?* One practical application of the policies learn with PGPSE, in line with the motivation of this work, is to utilize the experience gained during the exploration phase to construct informative datasets for offline RL (Yarats et al., 2022).

Starting from the datasets collected as for Figure 2, we relabel the transitions with sparse reward functions, assigning rewards of $+1$ for a specific goal state and $0$ otherwise. We then train an agent with offline Q-learning (Watkins & Dayan, 1992). At each training step, the agent samples a mini-batch of transitions $(s, a, r, s')$ from the dataset and updates its Q-value as $Q(s, a) \leftarrow Q(s, a) + \alpha \left[ r + \gamma \max_{a'} Q(s', a') - Q(s, a) \right]$ where $\alpha$ is the learning rate, $\gamma$ is the discount factor, and $\max_{a'} Q(s', a')$ represents the best estimated discounted future reward.

In Figure 3, we report the barplots of the corresponding offline RL success rate on each dataset relabeled, one for every possible goal state, ordering the results according to their respective success rate. Notably, a dataset collected with parallel agents allows to achieve the best offline RL performance across a large number of states, far greater than datasets collected with a single maximum entropy agent or a random policy. In Appendix C, we provide further heatmaps visualizations showing the goal positions that offline RL algorithm is able to reach (Figure 9), and how each dataset is looking in terms of their empirical state distribution (Figure 8).

**Takeaways.** Parallel State Entropy Maximization leads to better exploration, diverse data collection and improved performances in offline learning in PMDPs.

# 7. Related Work

In the following, we summarize the most relevant works this paper sets its root in, spanning from entropy maximization, multi-agent exploration to diverse skill discovery.

**State Entropy Maximization.** State entropy maximization in MDPs has been introduced in Hazan et al. (2019), from which followed a variety of subsequent works focusing on the problem from various perspectives (Lee et al., 2019; Mutti & Restelli, 2020; Mutti et al., 2021; 2022a;b; Mutti, 2023; Zhang et al., 2021a; Guo et al., 2021; Liu & Abbeel, 2021b;a; Seo et al., 2021; Yarats et al., 2021; Nedergaard & Cook, 2022; Yang & Spaan, 2023; Tiapkin et al., 2023; Jain et al., 2023; Kim et al., 2023; Zisselman et al., 2023; Zamboni et al., 2024a;b; 2025).

**Policy Gradient.** In the RL literature, first-order methods have been extensively employed to address non-concave policy optimization (Sutton et al., 1999; Peters & Schaal, 2008). In this work, we proposed a *vanilla* policy gradient estimator (Williams, 1992) as a first step, yet several further refinements could be made, such as natural gradient (Kakade, 2001), trust-region schemes (Schulman et al., 2015), and importance sampling (Metelli et al., 2018).

**Exploration with Multiple Agents.** The problem of exploration in the presence of multiple agents is a vast and diverse topic. Among others, concurrent exploration (Guo & Brunskill, 2015; Parisotto et al., 2019; Jung et al., 2020; Chen et al., 2022; Qin et al., 2022) has focused on scenarios where multiple agents operate within the same environment and learn simultaneously. Of particular interest are the works that investigate the role of information sharing between the agents (Alfredo & Arjun, 2017; Holen et al., 2023) or the ones that characterize the theoretical conditions for efficient coordinated exploration to happen and scale (Dimakopoulou & Van Roy, 2018; Dimakopoulou et al., 2018). Finally, exploration in Multi-Agent RL (MARL, Albrecht et al., 2024) has gained attention but was almost limited to reward-shaping techniques based on many heuristics Wang et al. (2019); Zhang et al. (2021b); Xu et al. (2024). In particular, Zhang et al. (2023) proposes a term maximizing the deviation from (jointly) explored regions. Yet, we highlight a key distinction: In conventional MARL, multiple agents operate concurrently within a shared environment, requiring coordination due to the interdependence of their trajectories; In contrast, our setting involves multiple agents interacting with independent instances of the environment. While coordination remains beneficial for enabling specialization and aligning agents with distinct objectives, their trajectories are decoupled. This structural difference fundamentally separates our framework from typical MARL formulations like Li et al. (2021), Zhao et al. (2021) and Lupu et al. (2021).

**Diverse Skills Discovery.** The problem of learning a set of diverse skills and/or policies is somehow linked to the problem of diversifying exploration via parallelization, and it has been addressed by a plethora of recent works (Hansen et al., 2019; Sharma et al., 2020; Campos et al., 2020; Liu & Abbeel, 2021a; He et al., 2022). The results provided by Gregor et al. (2017); Eysenbach et al. (2019) are particularly relevant for this work, in which the mutual information between visited states and skills is maximized, togheter with the ones in Zahavy et al. (2022), that explicitly maximizes the diversity between policies.

# 8. Conclusions

This paper targeted the critical challenge of efficient exploration in reinforcement learning, particularly in scenarios with slow environment simulators or limited interaction budgets. We introduced a novel parallel learning framework that leverages the power of multiple agents to maximize state entropy while explicitly promoting diversity in their exploration strategies. Unlike traditional approaches that rely on multiple instances of identical agents, our method, PGPSE based on policy gradient techniques, employs a centralized learning mechanism that balances individual agent entropy with inter-agent diversity, effectively minimizing redundancy and maximizing the information gained from parallel exploration.

Our theoretical analysis demonstrated the significant impact of parallelization on the rate of entropy stabilization and exploration diversity. Specifically, we showed that parallel agents, by focusing on different parts of the state space, can achieve faster convergence to high-entropy distributions compared to a single agent. These theoretical findings were strongly corroborated by our empirical results on various gridworld environments. PGPSE consistently outperformed single-agent baselines in terms of both state entropy and support size, demonstrating the effectiveness of our diversity-promoting approach.

Furthermore, we showed that the datasets collected by parallel agents exhibit higher entropy and lead to improved performance in offline settings, highlighting the practical value of our method for data-efficient learning. While our results are promising, we acknowledge certain limitations. Our current framework is evaluated primarily on discrete gridworld environments. Future work will involve extending PGPSE to continuous and more complex environments, such as those found in robotics and control problems.Future work will extend, the concept of state entropy maximization to the trajectory based one, with centrally, the benefit of parallelizing the exploration strategy to achieve wider and diverse skills acquisition over complicated real world problems. These extensions will be crucial for demonstrating the scalability of our approach to a broader use case.

## Impact Statement

This paper presents work whose goal is to advance the field of Machine Learning. There are many potential societal consequences of our work, none which we feel must be specifically highlighted here.

## Acknowledgment

This paper is supported by FAIR (Future Artificial Intelligence Research) project, funded by the NextGenerationEU program within the PNRR-PE-AI scheme (M4C2, Investment 1.3, Line on Artificial Intelligence). This work is supported by Siemens Digital Industry Italy.

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

# A. Main Proofs and Additional Results

**Concentration Properties.** In the following, we present the proofs of the theoretical results that highlight how the entropy of the distribution of state or trajectory visits concentrates in the single-agent and parallel-agent cases.

**Theorem A.1.** *Let $d_\pi$ be the (categorical) distribution induced by $\pi$ over the finite set $\mathcal{S}$ with $|\mathcal{S}| = S$, and let $d_n$ be the empirical distribution obtained from $n$ independent samples drawn from $d_\pi$. Then, for any $\epsilon > 0$, the following bound holds:*

$$\mathbb{P}\left(\mathcal{H}(d_\pi) - \mathcal{H}(d_n) > \epsilon\right) \leq 2S \exp\left(-n\frac{\epsilon^2 \mathrm{Var}(d_\pi)}{2S^3 \mathcal{H}^2(d_\pi)}\right),$$

*where $H(d_n)$ and $H(d_\pi)$ denote the entropy of the empirical and true distributions, respectively, and $\mathrm{Var}(d_\pi) = \sum_{s\in[\mathcal{S}]} d_\pi(s)(1 - d_\pi(s))$ is the variance of a random variable associated with the categorical distribution $d_\pi$. Furthermore, to ensure this concentration with confidence $1 - \delta$, the number of samples $n$ must satisfy the following lower bound:*

$$n \geq \frac{2S^3 \mathcal{H}^2(d_\pi)}{\epsilon^2 \mathrm{Var}(d_\pi)} \cdot \ln\frac{2S}{\delta}.$$

*Proof.* The proof consists of three main steps. In order to keep the derivation agnostic from the state or trajectory-based setting, we will now introduce a different yet equivalent notation: let $p$ be a categorical distribution over a finite set $\mathcal{X}$ with $|\mathcal{X}| = K$, and let $\hat{p}$ be the empirical distribution obtained from $n$.

**Decomposing the Problem via Union Bound.** First, we expand the entropy terms to highlight the contribution of the single components:

$$\mathbb{P}(\mathcal{H}(p) - \mathcal{H}(\hat{p}) > \epsilon) \leq \mathbb{P}\left(\sum_{i=1}^K p_i \log\left(\frac{1}{p_i}\right) - \hat{p}_i \log\left(\frac{1}{\hat{p}_i}\right) > \epsilon\right) = \mathbb{P}\left(\sum_{i=1}^K h(p_i) - h(\hat{p}_i) > \epsilon\right),$$

where $p_i = \mathbb{P}(X = x_i)$ and $h(x) = x\log\left(\frac{1}{x}\right)$.

Applying the union bound to the previous result, we get:

$$\mathbb{P}(\mathcal{H}(p) - \mathcal{H}(\hat{p}) > \epsilon) \leq \sum_{i=1}^K \mathbb{P}\left(h(p_i) - h(\hat{p}_i) > \frac{\epsilon}{K}\right). \tag{2}$$

**Bounding the Entropy of the Components using a Linear Approximation.** Now, we focus on finding an upper bound to $h(p_i) - h(\hat{p}_i)$. We introduce a lower bound to $h(\hat{p}_i)$ obtained by a combination of functions that are linear in the deviation $|p_i - \hat{p}_i|$:

$$h(\hat{p}_i) \geq h(p_i) - \frac{h(p_i)|p_i - \hat{p}_i|}{\min(p_i, 1 - p_i)} \geq h(p_i) - \frac{h(p_i)|p_i - \hat{p}_i|}{p_i(1 - p_i)}.$$

As a consequence

$$\mathbb{P}\left(h(p_i) - h(\hat{p}_i) > \epsilon\right) \leq \mathbb{P}\left(\frac{h(p_i)|p_i - \hat{p}_i|}{p_i(1 - p_i)} > \epsilon\right) \leq \mathbb{P}\left(|p_i - \hat{p}_i| > \frac{p_i(1 - p_i)}{h(p_i)}\epsilon\right). \tag{3}$$

Thanks to this last inequality, we can now focus on the concentration inequality of the Bernoulli distributions associated with the parameters $p_i$.

**Applying a Concentration Inequality for Bernoulli Distributions.** Finally, we use a concentration inequality on the estimation of a Bernoulli-distributed parameter to express this probability bound in terms of the variance of $p_i$ ($\mathrm{Var}(p_i) = p_i(1 - p_i)$).

Leveraging Chernoff bound for Bernoulli distributions, we get:

$$\mathbb{P}(|p_i - \hat{p}_i| > \epsilon) \leq e^{-nD_{KL}(p_i+\epsilon\|p_i)} + e^{-nD_{KL}(p_i-\epsilon\|p_i)} \leq 2e^{-\frac{n\epsilon^2}{2p_i(1-p_i)}} = 2e^{-\frac{n\epsilon^2}{2\mathrm{Var}(p_i)}}. \tag{4}$$

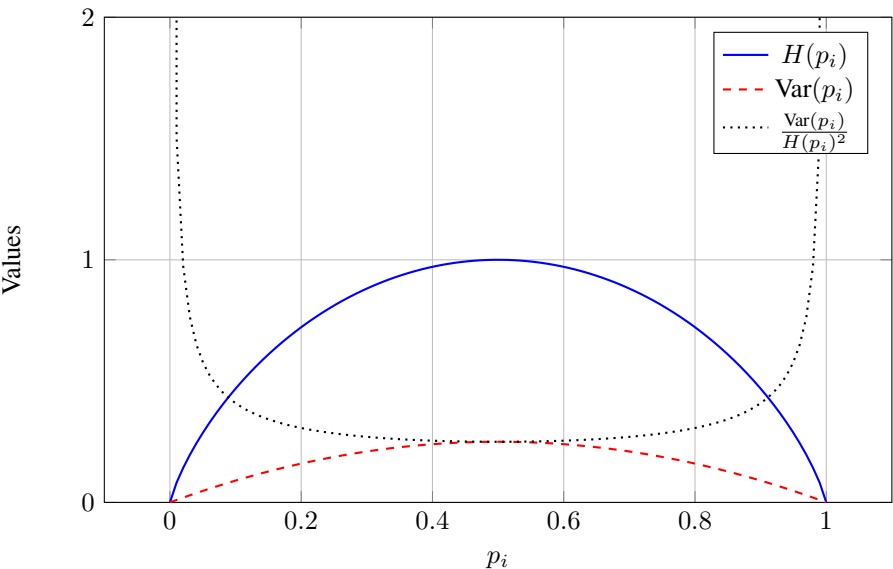

*Figure 4.* The plot shows that $H(p_i)$ and $\mathrm{Var}(p_i)$ are concave symmetric function with their maximum located at $p_i = 0.5$, while $\frac{\mathrm{Var}(p_i)}{H(p_i)^2}$ is a convex symmetric function with its minimum located at $p_i = 0.5$.

We now complete the proof by combining the results in Eqs (2), (3), and (4):

$$\mathbb{P}(\mathcal{H}(p) - \mathcal{H}(\hat{p}) > \epsilon) \leq \sum_{i=1}^{K} \mathbb{P}\left(h(p_i) - h(\hat{p}_i) > \frac{\epsilon}{K}\right) \leq \sum_{i=1}^{K} \mathbb{P}\left(|p_i - \hat{p}_i| > \frac{p_i(1 - p_i)}{Kh(p_i)}\epsilon\right) \leq 2\sum_{i=1}^{K} e^{-\frac{n\epsilon^2 p_i(1 - p_i)}{2K^2 h^2(p_i)}}. \quad (5)$$

In order to remove the summation over the $K$ components of the distribution, we need to find a lower bound to the term $\frac{p_i(1-p_i)}{\mathcal{H}^2(p_i)}$ that is independent of the specific component parameter $p_i$. Here, we show the chain of passages that achieve this goal:

$$\min_i \frac{p_i(1 - p_i)}{h^2(p_i)} \geq \min_i \frac{p_i(1 - p_i)}{\mathcal{H}^2(p_i)} = \frac{\max_i p_i(1 - p_i)}{\max_i \mathcal{H}^2(p_i)} \geq \frac{\sum_i p_i(1 - p_i)}{K \max_i \mathcal{H}^2(p_i)} \geq \frac{\mathrm{Var}(p)}{K\mathcal{H}^2(p)}.$$

The motivations for each step are:

1. $\mathcal{H}(p_i) \geq h(p_i)$.

2. The value of $p_i$ that minimizes $\frac{p_i(1-p_i)}{\mathcal{H}^2(p_i)}$ is the one with the highest entropy (see Figure4). The ration between the variance $p(1 - p)$ and the squared entropy $\mathcal{H}^2(p)$ is symmetric about $p = \frac{1}{2}$ and it has negative derivative for $p < \frac{1}{2}$ and positive derivative for $p > \frac{1}{2}$. Since the higher the entropy $\mathcal{H}(p_i)$, the higher is also the variance $p_i(1 - p_i)$, we can restate the minimization problem as the ratio of two maximization problems.

3. The term in the numerator is the maximum variance, which can be lower bounded by the average variance.

4. The maximum entropy among the Bernoulli distributions associated with all the components is upper bounded by the entropy of the categorical distribution $p$.

Leveraging this result in Eq. (5) concludes the proof.

$\square$

**Frank-Wolfe for Parallel Exploration with Infinite Trials** Frank-Wolfe Algorithms have gained much attention for their strong theoretical guarantees in MDPs (Hansen et al., 2019). In this section, we show that by performing Frank-Wolfe-like updates in a parallel fashion over *infinite-trials* objectives as reported in Algorithm 2, it is possible to obtain *faster convergence rates* with respect to the non-parallel formulation, through a convenient scaling factor of $1/N$ with $N$ being the number of parallel instantiations. In order to allow for a simpler derivation, we will assume access to two kinds of oracles. First, some **approximate planning oracles** (one per each agent) that given a reward function (on states) $r : S \to \mathbb{R}$ and a sub-optimality gap $\epsilon_1$, returns a policy $\pi = \text{APPROXPLAN}(r, \epsilon_1)$ with the guarantee that $\mathcal{H}(\pi) \geq \max_{\bar{\pi}} \mathcal{H}(\bar{\pi}) - \epsilon_1$. In addition, some **state distribution estimate oracles** (one per each agent) that estimate the state distribution $\hat{d} = \text{DENSITYEST}(\pi, \epsilon_0)$ of any given (non-stationary) policy $\pi$, guaranteeing that $\|d^\pi - \hat{d}\|_\infty \leq \epsilon_1$. In addition, we will assume that the entropy functional $\mathcal{H}$ is $\beta$-smooth, $B$-bounded, and that it satisfies the following inequality for all $X, Y$:

$$\|\nabla \mathcal{H}(X) - \nabla \mathcal{H}(Y)\| \leq \beta \|X - Y\|_\infty$$
$$-\beta \mathbb{I} \leq \nabla^2 \mathcal{H}(X) \leq \beta \mathbb{I}; \|\nabla \mathcal{H}(X)\|_\infty \leq B$$

Under these assumptions, it follows that Algorithm 2 enjoys the following:

**Theorem A.2** (Convergence Rate). *For any $\varepsilon > 0$, set $\varepsilon_1 = 0.1\varepsilon$, $\varepsilon_0 = 0.1\beta^{-1}\varepsilon$, and $\eta = 0.1|\mathcal{S}|^{-1}\beta^{-1}N\varepsilon$, where Algorithm 2 is run for $T$ iterations over $N$ agents in parallel where:*

$$T \geq 10\beta|\mathcal{S}|N^{-1}\varepsilon^{-1} \log 10B\varepsilon^{-1},$$

*we have that:*

$$H(\pi_{mix,T}) \geq \max_\pi H(d_\pi) - \varepsilon.$$

*Proof of Theorem A.2.* Let $\pi^*$ be a maximum-entropy policy, ie. $\pi^* \in \text{argmax}_\pi H(d_\pi)$.

$$H(d_{\pi_{\text{mix},t+1}}) = H((1 - \eta)d_{\pi_{\text{mix},t}} + \eta d_{\pi_{t+1}})$$
$$\geq H(d_{\pi_{\text{mix},t}}) + \eta \langle d_{\pi_{t+1}} - d_{\pi_{\text{mix},t}}, \nabla H(d_{\pi_{\text{mix},t}})\rangle - \eta^2 \beta \|d_{\pi_{t+1}} - d_{\pi_{\text{mix},t}}\|_2^2$$
$$\geq H(d_{\pi_{\text{mix},t}}) + \frac{\eta}{N} \sum_i \langle d_{\pi_{t+1}^i} - d_{\pi_{\text{mix},t}}, \nabla H(d_{\pi_{\text{mix},t}})\rangle - \frac{\eta^2 \beta}{N^2} \sum_i \|d_{\pi_{t+1}^i} - d_{\pi_{\text{mix},t}}\|_2^2$$

The second inequality follows from the smoothness of $H$, the third applies the definition of distributions induced by mixture policies. Now, to incorporate the error due to the two oracles, observe that for each agent it holds

$$\langle d_{\pi_{t+1}^i}, \nabla H(d_{\pi_{\text{mix},t}})\rangle \geq \langle d_{\pi_{t+1}^i}, \nabla H(\hat{d}^i_{\pi_{\text{mix},t}})\rangle - \beta \|d_{\pi_{\text{mix},t}} - \hat{d}^i_{\pi_{\text{mix},t}}\|_\infty$$
$$\geq \langle d_{\pi^*}, \nabla H(\hat{d}^i_{\pi_{\text{mix},t}})\rangle - \beta \varepsilon_0 - \varepsilon_1$$
$$\geq \langle d_{\pi^*}, \nabla H(d_{\pi_{\text{mix},t}})\rangle - 2\beta \varepsilon_0 - \varepsilon_1$$

The first and last inequalities invoke the assumptions on the entropy functional. Note that the second inequality above follows from the defining character of the planning oracle. Using the above fact and continuing on

$$H(d_{\pi_{\text{mix},t+1}}) \geq H(d_{\pi_{\text{mix},t}}) + \frac{\eta}{N} \sum_i \langle d_{\pi_{t+1}^i} - d_{\pi_{\text{mix},t}}, \nabla H(d_{\pi_{\text{mix},t}})\rangle - \frac{\eta^2 \beta}{N^2} \sum_i \|d_{\pi_{t+1}^i} - d_{\pi_{\text{mix},t}}\|_2^2$$

$$\geq H(d_{\pi_{\text{mix},t}}) + \eta \langle d_{\pi^*} - d_{\pi_{\text{mix},t}}, \nabla H(d_{\pi_{\text{mix},t}})\rangle - 2\beta\eta\varepsilon_0 - \eta\varepsilon_1 - \frac{\eta^2 \beta}{N}|\mathcal{S}|$$

$$\geq (1 - \eta)H(d_{\pi_{\text{mix},t}}) + \eta H(d_{\pi^*}) - 2\eta\beta\varepsilon_0 - \eta\varepsilon_1 - \frac{\eta^2 \beta|\mathcal{S}|}{N}$$

The last step here utilizes the concavity of $H$. It follows that

$$H(d_{\pi^*}) - H(d_{\pi_{\text{mix},t+1}}) \leq (1 - \eta)(H(d_{\pi^*}) - H(d_{\pi_{\text{mix},t}})) + 2\eta\beta\varepsilon_0 + \eta\varepsilon_1 + \frac{\eta^2\beta|\mathcal{S}|}{N}.$$

Telescoping the inequality, this simplifies to

$$H(d_{\pi^*}) - H(d_{\pi_{\text{mix},T}}) \leq (1 - \eta)^T(H(d_{\pi^*}) - H(d_{\pi_{\text{mix},0}})) + 2\beta\varepsilon_0 + \varepsilon_1 + \eta\beta$$
$$\leq Be^{-T\eta} + 2\beta\varepsilon_0 + \varepsilon_1 + \frac{\eta^2\beta|\mathcal{S}|}{N}.$$

$$H(d_{\pi^*}) - H(d_{\pi_{\text{mix},T}})$$
$$\leq (1 - \eta)^T(H(d_{\pi^*}) - H(d_{\pi_{\text{mix},0}})) + 2\beta\varepsilon_0 + \varepsilon_1 + \frac{\eta\beta|\mathcal{S}|}{N}$$
$$\leq Be^{-T\eta} + 2\beta\varepsilon_0 + \varepsilon_1 + \frac{\eta\beta|\mathcal{S}|}{N}.$$

Setting $\varepsilon_1 = 0.1\varepsilon$, $\varepsilon_0 = 0.1\beta^{-1}\varepsilon$, $\eta = 0.1N|\mathcal{S}|^{-1}\beta^{-1}\varepsilon$, $T = \eta^{-1}\log 10B\varepsilon^{-1}$ leads to the final result. $\qquad\square$

---

**Algorithm 2** Parallel Frank-Wolfe.

---

1: **Input:** Step size $\eta$, number of iterations $T$, number of agents $N$, planning oracle tolerance $\varepsilon_1 > 0$, distribution estimation oracle tolerance $\varepsilon_0 > 0$.
2: Set $\{C_0^i = \{\pi_0^i\}\}_{i \in N}$ where $\pi_0^i$ is an arbitrary policy, $\alpha_0^i = 1$.
3: **for** $t = 0, \ldots, T - 1$ **do**
4:     Each agent call the state distribution oracle on $\pi_{\text{mix},t} = \frac{1}{N}\sum_i(\alpha_t^i, C_t^i)$:

$$\hat{d}_{\pi_{\text{mix},t}}^i = \text{DENSITYEST}\left(\pi_{\text{mix},t}, \varepsilon_0\right)$$

5:     Define the reward function $r_t^i$ for each agent $i$ as

$$r_t^i(s) = \nabla H(\hat{d}_{\pi_{\text{mix},t}}^i) := \left.\frac{d\mathcal{H}(X)}{dX}\right|_{X=\hat{d}_{\pi_{\text{mix},t}}^i}.$$

6:     Each agent computes the (approximately) optimal policy on $r_t$:

$$\pi_{t+1}^i = \text{APPROXPLAN}\left(r_t^i, \varepsilon_1\right).$$

7:     Each agent updates

$$C_{t+1}^i = (\pi_0^i, \ldots, \pi_t^i, \pi_{t+1}^i), \tag{6}$$
$$\alpha_{t+1}^i = ((1 - \eta)\alpha_t^i, \eta). \tag{7}$$

8: **end for**
9: $\pi_{\text{mix},T} = \frac{1}{N}\sum_i(\alpha_T^i, C_T^i)$.

---

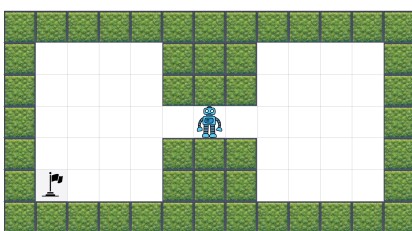

*Figure 5.* Two rooms gridworld, with starting position in the corridor and goal state in the left bottom corner.

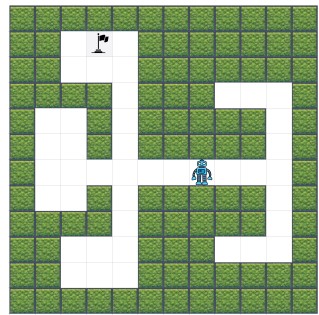

*Figure 6.* Maze gridworld, with multiple path, goal state in the upper section

## B. Experimental Details

**Environments.** In this paper, we present two distinct grid-world environments designed to illustrate the effectiveness of parallel exploration in enabling agents to overcome exploration bottlenecks and efficiently discover optimal trajectories in both structured and intricate scenarios.. In the first, two rooms are connected through a corridor with the goal positioned in the left room. In the second one, we considered a maze-like space, with a more complex layout with many bifurcations but only one path leading to a goal. The two environments are shown in Figures 5 and 6. In the following, we briefly describe the two environments more thorough.

### B.1. Room

This is a $5 \times 11$ grid world inspired by the toy example from Towers et al. (2024). It consists of two rooms connected by a single horizontal corridor. The agent moves until reaching the goal or the episode terminates. The starting position is fixed at the center of the corridor, $(2, 5)$. The agent navigates the grid using actions from $\{0, 1, 2, 3\}$, where the index indicate $0$: *Move left*, $1$: *Move down*, $2$: *Move right*, $3$: *Move up*. The observation space represents the player's current position as an index computed by obs $=$ row $\times$ ncols $+$ col where rows and columns are zero-indexed. For instance, the goal position $(4, 0)$ is mapped to $4 \times 11 + 0 = 44$.. The total number of observations depends on the grid size. The environment is evaluated in both *reward-free* and *sparse-reward* settings, where in the latter case reward function $r : S \times A \to \mathbb{R}$ with $r = 1$ upon reaching the goal and $0$ otherwise. Throughout the paper, we consider two configurations:

- *Room-det*: A deterministic version where executing an action $a$ under policy $\pi_\theta$ always results in the intended movement.

- *Room-stoc*: A stochastic variant where the agent plays the intended action with probability $1 - p$ and deviate from it with $p$.

### B.2. Maze

This is a grid world $10 \times 10$. The environment consists of a discrete grid-based navigation task, where an agent must reach a goal position while navigating through a structured maze. The grid is structured to restricting movement to specific corridors. The agent, starts from an initial position $(5, 6)$ and can move in four directions: up, down, left, or right, as in the B.1 case. The goal serves as the terminal state, where the episode ends upon successful navigation or after a number of maximum steps. Upon a different structure, the environment shares with the previous environment the reward function, the action space and the state index conversion. In the paper we present two different version of the environment as following defined:

- *Maze-det*: A deterministic version where executing an action $a$ under policy $\pi_\theta$ always results in the intended movement.

- *Maze-stoc*: A stochastic variant where the agent plays the intended action with probability $1 - p$ and deviate from it with $p$.

**Implementation Details of Algorithm 1.** As outlined in the pseudocode of Algorithm 1, in each episode, a batch $|\mathcal{B}| = 40$, of $K$ trajectories are experienced by parallel learners. $K$ is fixed during the experiment always to $1$, due to

the assumption of having access to the minimum size of possible interactions that the agents can have with respect to the environment. To reproduce the policy gradient performance of the single agent, this $K$ should coincide to the number of agent $m$, at which the single is compared to. Indeed, in the 1 with respectively $\{2, 6\}$ agents involved, the single agent experiences $K = \{2, 6\}$ trajectories, from which the state distribution $d_n$ is derived. Over the $10k$ episodes, the parallel agents interact with the environment with finite trajectories of lenght $T$ that in the *Room-det* and *Room-stoc* environments is set to 8, while in *Maze-det* and *Maze-stoc*, due to a bigger state space, is set to 10.

The parallel policy, as collection of parametric single policy on $\theta$, is a Softmax function defined as:

$$\pi_\theta(a|s) = \frac{e^{\theta \cdot s}}{\sum_{a' \in \mathcal{A}} e^{\theta \cdot s}}$$

$\theta$ parameters are updated via gradient ascent maximization, with learning rate $\alpha$ set initially to 0.1 with exponential decay rate $\lambda$ of 0.999. The training is performed over 5 different seeds $\{0, 1, 2, 42, 133\}$.

## C. Further Experimental Details and Analysis

The code to reproduce the experiments is made available at the following link.

### C.1. Performance of Four Parallel Agents

To further validate our approach, we extend the experimental analysis presented in the main body by including results for four parallel agents. Figure 7 illustrates the performance of this configuration in terms of normalized state entropy (top row) and support size (bottom row) across the four environments: *Room-det*, *Room-stoc*, *Maze-det*, and *Maze-stoc*.

As observed, the four-agent configuration consistently outperforms the single-agent baseline, achieving significantly higher normalized entropy and support size. This demonstrates the continued effectiveness of our parallelized exploration strategy, with the centralized update mechanism of Algorithm 1 enabling the agents to collectively learn policies that lead to a more diverse and comprehensive exploration of the state space.

However, it is also evident that the marginal benefit of adding more agents diminishes as the number of agents increases. This phenomenon is likely attributable to the relatively small size of the state space in the considered environments. With more agents, the overlap in their explored trajectories increases, leading to a less pronounced improvement for each additional agent.

### C.2. Dataset Analysis

To further understand the benefits of parallel exploration, we analyze the datasets generated by the learned parametric policies. These datasets are constructed analogously to those in the single-agent case and the random policy baseline. Specifically, for each parallel-agent configuration, we sample $K = m$ trajectories to ensure a consistent comparison across settings. The mean and standard deviation of the state entropy are computed over these trajectories and averaged across the five random seeds used during training. Let's look deeply at the dataset created from the paralell agents, when interacting with the more complex *Maze* environment. Figures 8 present heatmaps visualizing the state occupancy distributions for the *Maze* environments, respectively the deterministic and stochastic case. These heatmaps provide a qualitative comparison of the exploration patterns achieved by single agents, parallel agents, in the two implemented versions.

### C.3. Offline Q-Learning Implementation

While single agents might discover effective exploration strategies, it is the diversity inherent to our parallel policy induction that ensures robust and consistent state space coverage, even in stochastic environments. This diversity, validated across multiple random seeds, translates into a highly informative dataset for downstream learning tasks. The effectiveness of our approach is further highlighted by the fact that this valuable dataset is constructed from a mere six trajectories per agent, a significant improvement over the data generated by untrained policies. Combined with an offline procedure, the paralell approach incredibly speedup the learning procedure. For the construction of Figure 3, we implement an offline Q-learning algorithm. The agent learns from a replay buffer with a batch size of 20 $(s, a, s', r)$ interactions. The experiences collected from the parallel agents are incorporated into the replay buffer, without interacting with the environment. The Q-learning agent is trained for 100 episodes, with the Q-value function $Q(s, a)$ being updated using a learning rate of $\alpha = 0.1$ and a

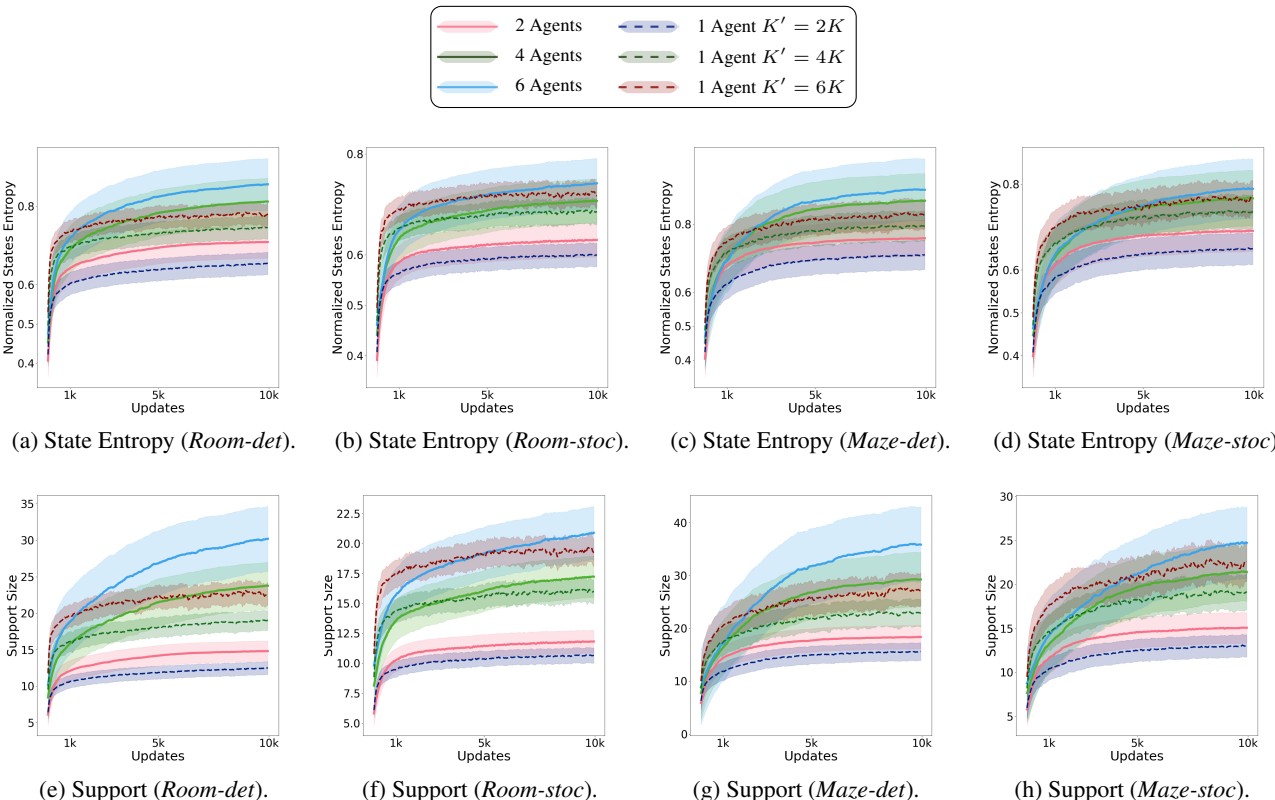

*Figure 7.* Performance of four parallel agents. The top row (a–d) shows the progression of normalized state entropy across different environments, while the bottom row (e–h) depicts the corresponding support size dynamics. Each plot illustrates the performance of parallel agents against a single instance visiting a number of trajectories corresponding to the number of parallel agents.

discount factor of $\gamma = 0.99$ according to the Bellman equation update rule.

The high quality of the dataset generated by our parallel agents is further evidenced in Figure 9. The offline agent, trained solely on this data, demonstrates significantly improved goal-reaching capabilities, even in the stochastic environment, as indicated by the broader and warmer-colored regions of the heatmap, particularly for goals distant from the starting location.

### C.4. Diversity Analysis

Figure 10 provides a visual comparison of the learned policies for parallel agents versus single agents in both *Room-det* and *Maze-det* environments. In the single-agent setting (Figures 10a and 10b), the agent tends to develop policies with higher stochasticity in its action choices. This is a consequence of the need to explore a larger portion of the state space alone, necessitating a more randomized approach to discover high-entropy trajectories. In contrast, the parallel agent setting (Figures 10c and 10d) reveals that the centralized update mechanism, based on the aggregate state distribution $d_n$, encourages individual agents to specialize in more deterministic policies.

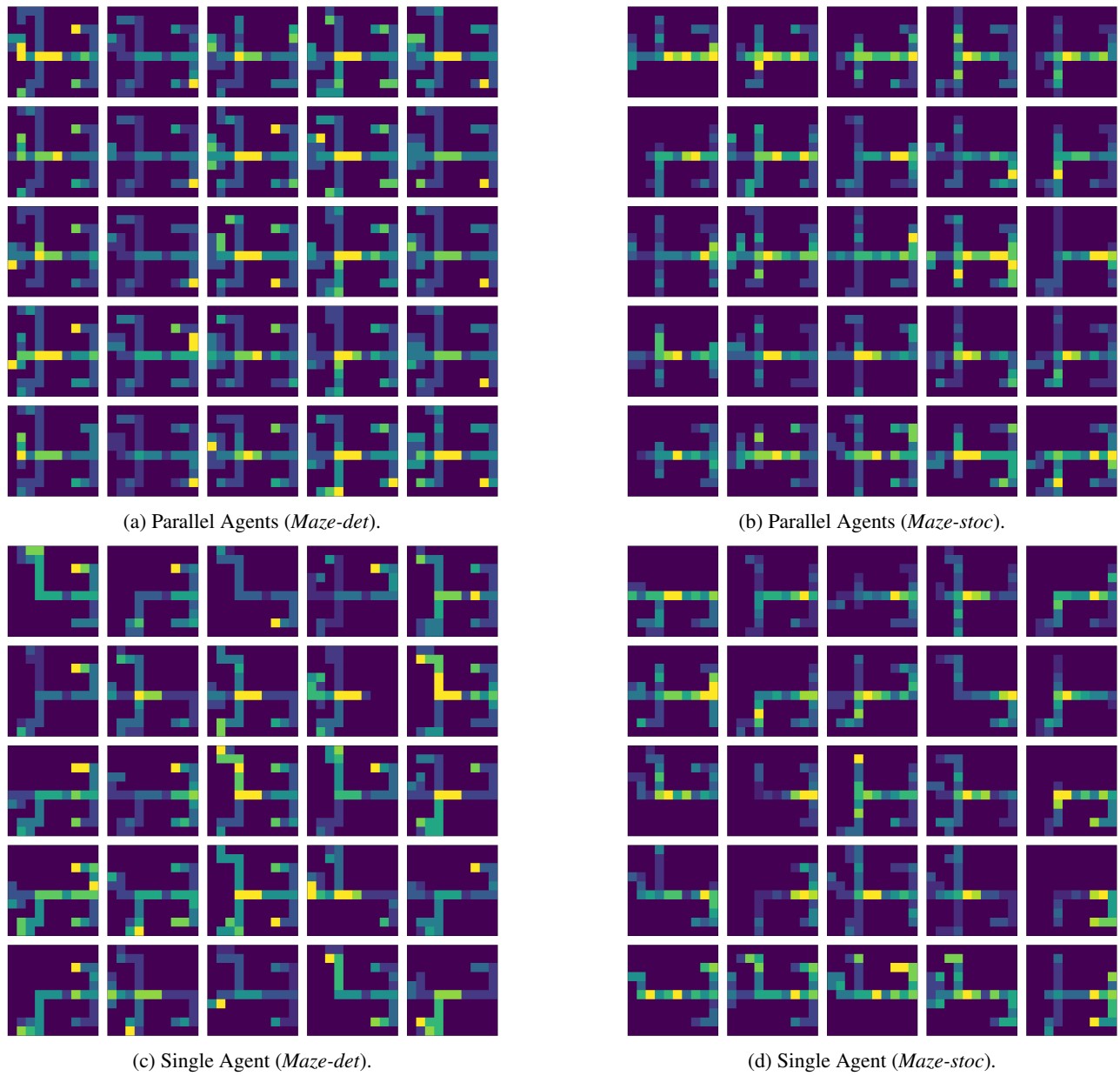

(a) Parallel Agents (*Maze-det*).

(b) Parallel Agents (*Maze-stoc*).

(c) Single Agent (*Maze-det*).

(d) Single Agent (*Maze-stoc*).

*Figure 8.* Dataset extrapolation based on the *Maze* environments.

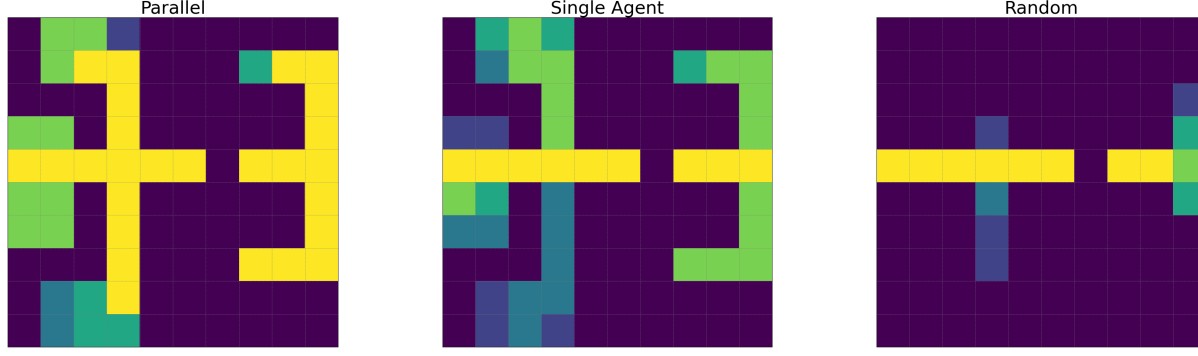

(a) Offline Goal Reachability, learning from the parallel agents experience.

(b) Offline Goal Reachability, learning from the single agents experience.

(c) Offline Goal Reachability, learning from a random policy experience.

*Figure 9.* Offline Result of Gained Experience

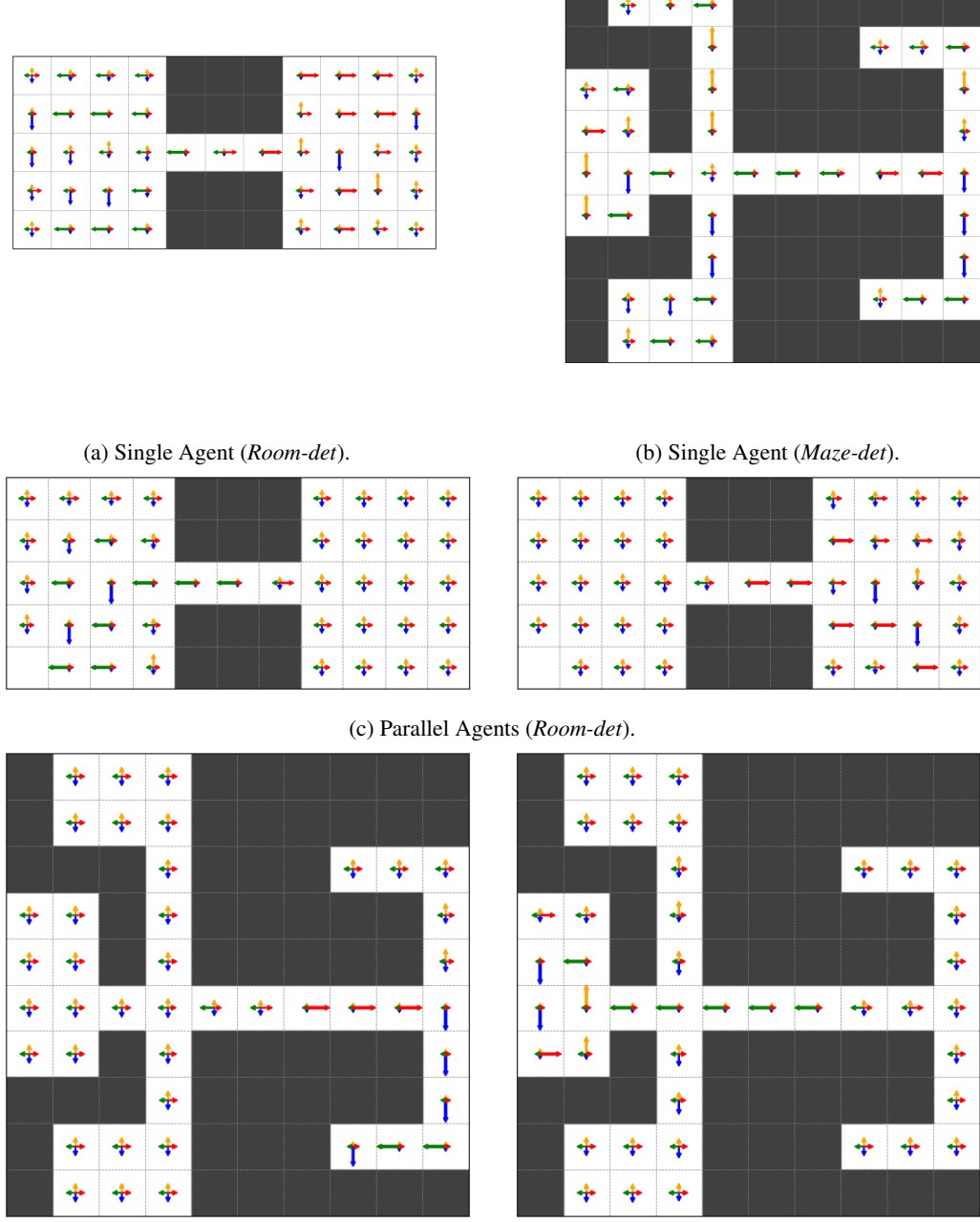

(a) Single Agent (*Room-det*).

(b) Single Agent (*Maze-det*).

(c) Parallel Agents (*Room-det*).

(d) Parallel Agents (*Maze-det*).

*Figure 10.* Analysis of stochasticity and determinism of parallel vs. single agents.

