# OpenReview forum: "Enhancing Diversity In Parallel Agents: A Maximum State Entropy Exploration Story"
_ICML.cc/2025/Conference — ICML 2025 poster_

### Official Review · Reviewer_khWs · 2025-03-13

**Overall Recommendation:** 3

**Summary:**

This paper focuses on generating diverse experience for policy gradient algorithms in reward-free settings through the use of entropy maximisation and separate parallel policies. The method proposed is Policy Gradient for Parallel States Entropy Maximization (PGPSE). The empirical results on two grid-based environments show that more diverse policies lead to a higher entropy of collected states which corroborates the theoretical findings.

**Claims And Evidence:**

While there is theoretical evidence for this method, the empirical evidence is severely lacking for multiple reasons:

1. In section 5 the authors state: “As stated before, a core motivation of this work is addressing the problem of exploration in practical scenarios, where a strong parallelization of the environment is used to overcome the hardness to sample from complex simulators or even physical instantiations of the agents.”  If a core motivation of this work is practical applications, then why are the environments so simple? They only have 43 states, which is significantly simpler than the common benchmark environments used in similar work [mujoco,atari,gym].

2. Why was training only shown for 2, 4 and 6 agents? Especially since a higher K’ leads to higher state entropy and larger support size in Figure 1 (for stochastic environments). It would be interesting to see the trend of the gap between single agent and multi-agent as the number of agents grows.

3. I would like to see a small discussion around the higher variance introduced with more agents. Especially in the stochastic environments, as it seems more agents may not be statistically significantly better than a single agent in those environments given the high variance.

**Essential References Not Discussed:**

Not that I am aware of.

**Experimental Designs Or Analyses:**

Yes, experimental design is valid, except for the fact that the environments are simple.

**Methods And Evaluation Criteria:**

Not comparing to other baselines makes it hard to place this work within the literature. For example, Hazan, Elad, et al [maxent] compared to a random policy, it would have been interesting to compare to MaxEnt and a random policy. Another simple baseline that seems to have been left out is N distinct agents learning without the entropy objective.

Additionally, while multiple seeds were used for this work, I’d suggest that following the advice of Argarwal et al. [rliable] and doing a more rigorous evaluation.

**Other Comments Or Suggestions:**

The paper is poorly written, containing numerous grammatical and formatting errors. I cannot enumerate them all here, but here are a few examples:
1. Figures 2 and 3 lack labels for what the colours are referring to.
2. Page 1 second last paragraph: performances -> performance
3. Page 4 section 4: “Main Takeout” -> “Main Takeaway”
4. Page 4 section 5: “hardness” -> “difficulty”
5. Page 4 section 5: “our attention now shift” -> “our attention now shifts”
6. Page 7 section 6 (Offline RL): “Can dataset collected with parallel maximum entropy agents benefit offline RL algorithms on those data?” -> “Can datasets collected with parallel maximum entropy agents benefit offline RL algorithms”

-----

**References**

[mujoco] Todorov, Emanuel, Tom Erez, and Yuval Tassa. "Mujoco: A physics engine for model-based control." 2012 IEEE/RSJ international conference on intelligent robots and systems. IEEE, 2012.
[atari] Bellemare, Marc G., et al. "The arcade learning environment: An evaluation platform for general agents." Journal of artificial intelligence research 47 (2013): 253-279.
[gym] Brockman, Greg, et al. "Openai gym." arXiv preprint arXiv:1606.01540 (2016).
[maxent] Hazan, Elad, et al. "Provably efficient maximum entropy exploration." International Conference on Machine Learning. PMLR, 2019.
[rliable] Agarwal, Rishabh, et al. "Deep reinforcement learning at the edge of the statistical precipice." Advances in neural information processing systems 34 (2021): 29304-29320.

**Other Strengths And Weaknesses:**

**Strengths:** I think the application to offline RL is very interesting and should be further emphasised throughout the paper and tested on harder problems.

**Weaknesses:** I feel the significance of this work is lacking. The authors mention a large amount of potential future research and I believe that this would need to be included for this paper to be considered for ICML.. As it stands, the environments tested are very simple and the paper lacks baselines, which makes it impossible to place in the literature. Additionally, the clarity of the writing is poor  and is a significant weakness of this work.

**Questions For Authors:**

Is there a reason for why the gap in state entropy seems to decrease as you increase the number of agents? In Figure 1, it seems that the gap between 6 agents and K’ = 6 is lower than 2 agents and K’ = 2.

**Relation To Broader Scientific Literature:**

It is related to a wealth of exploration literature and reward-free reinforcement learning, specifically those focusing on maximum entropy learning. It is also related to policy gradient methods in general, as in most cases rollouts are done in parallel, and thus this method could be applied.

**Theoretical Claims:**

I did not check the correctness of the proofs.

---

> ### Author Rebuttal · Authors · 2025-03-31
>
> We sincerely appreciate the reviewer's time and effort in evaluating our work, and we are grateful for the opportunity to provide further clarifications. We also thank the reviewer for highlighting grammatical and formatting issues, which we will address in the final version. Additionally, we are pleased that our contribution on applying the exploration approach to offline RL was appreciated.
>
> **Weaknesses: I feel the significance of this work is lacking.**
>
> We hear the reviewer's concerns about the simple domains and baselines. We will address them in detail below. First, we want to spend a few words to get on the same page with the reviewer on the nature of this work. The core contributions of this paper are *conceptual* and *theoretical*: The formulation of a reward-free exploration objective tailored to the parallel MDP setting (Section 3), its theoretical characterization through novel concentration bounds (Section 4 and Theorem 4.1), the design of a specialized policy gradient procedure (Section 5). While we corroborate the latter contributions with a *preliminary* empirical analysis, this is not meant to close the gap between foundations and application in the real world, which we believe requires a substantial amount of additional work for not just a follow-up but a series of papers. We kindly ask the reviewer to also weigh our main contributions in their evaluation and to reward the paper for its seminal potential.
>
> **If a core motivation of this work is practical applications, then why are the environments so simple?**
>
> Please note that we do not claim the paper provides a solution for real-world applications, but that this research line is *motivated* by real-world applications. We do not aim to close the gap with applications with our work, which, being just the first in this direction, aims to introduce the problem formally and to provide the theoretical foundations for future works.
>
> **Why was training only shown for 2, 4, and 6 agents?**
>
> Given the gridworld’s complexity, a few agents are sufficient for successful navigation, and adding more does not significantly improve performance due to state space constraints. In experiments on more complex scenarios, we expect a much larger number of learners to be beneficial for the exploration. For completeness, we will extend Figure 7 in the Appendix to show performance with more agents, highlighting how increasing their number maximize the objective until reaching a plateau.
>
> **I would like to see a small discussion around the higher variance introduced with more agents.**
>
> Variance in performance is an important aspect, which we will discuss further in the paper. While a single agent with more trajectories can achieve similar expected performance, parallel learners naturally exhibit smaller variance around the mean. This supported by Theorem 4.1 and the experiments in Figures 2 and 3 highlight the benefits of parallel training. A single agent tends to explore the entire state space, leading to higher variance in state distribution and greater sample complexity compared to the parallel case, in which the policies are less stochastic.
>
> **Is there a reason why the gap in state entropy decreases as the number of agents increases?**
>
> In designing the experiments, we aimed for a fair comparison. Comparing parallel agents, each playing one trajectory at a time against a single agent with the same interaction budget, would overestimate the parallel learners' advantage. Instead, we chose a more challenging setup where the single agent has a significantly larger interaction budget than each individual parallel agent. This setup allows the single agent to develop a stronger policy, reducing the gap in state entropy. However, when considering the offline RL setting, the advantage of parallel learning becomes more evident due to the higher variance induced by the policy of the single agent that need to visit more states, respect to a specialized parallel policy that concentrate faster.
>
> **Baselines makes it hard to place this work within the literature.**
>
> As our paper is the first to address this specific problem of state entropy maximization in parallel MDPs, we do not have a direct baseline with which to compare. However, we did compare against a *random policy* (gray bar in Figure 3 and 4) and a *single policy*, which can be seen as replicas of a MaxEnt algorithm across the parallel MDPs (the objective is the same as MaxEnt, although we used our implementation instead of the one of the original paper). Those are the baselines that the reviewer also mentioned, and we believe they are the most relevant. We will make this information clearer in the paper.

---

> > ### Comment · Reviewer_khWs · 2025-04-03
> >
> > I thank the authors for taking the time to answer my questions. They have cleared up my misunderstanding around practical applications and I see that my recommended baselines, although unlabeled, were already present in the paper. I also appreciate the clarity around the higher variance with more agents.
> >
> > >few agents are sufficient for successful navigation
> >
> > Please mention this in the paper. The reason I asked for this is not only to see how well PGPSE scales, but rather to see how well a single agent approach could do given more transitions as it seems that this performance scales well with $K$.
> >
> > >we did compare against a random policy (gray bar in Figure 3 and 4)
> >
> > Thank you for pointing this out, but given that figure 3 is unlabeled, how is a reader supposed to tell that the gray bar is a random agent? Additionally, it would be interesting to see the random agent curves in Figure 2.
> >
> > >single policy is equivalent to replicas of MaxEnt
> >
> > Could you please make this more clear in the experiment section?
> >
> > >The core contributions of this paper are conceptual and theoretical
> >
> > While I acknowledge that the theoretical proofs are a core contribution of this paper, it does not absolve the authors from performing adequate experiments to validate their proposed method, which is also listed as a core contribution in the introduction.
> >
> > I do not feel that what I am asking is unreasonable as I see other works in this field which are able to have both extensive proofs and experiments on complex environments [1,2]. In fact, some of these works even label experiments using robots simulated in MuJoCo (a much more complex environment than the ones present in this paper) as “preliminary” [1]. Some of these similar works also use a maze environment as done in this paper, but explicitly label them as “simple” environments or use them to easily display learned behaviours, but not as the main benchmark [2,3].
> >
> > To increase my score, ideally I want to see experiments on more complex environments aligned with the literature, however I realise this may be infeasible in such a small amount of time. Thus, I may be willing to increase my score if the significance of the current experiments are significantly downplayed, for example they should be presented as "preliminary" in the introduction, experiments and conclusion sections. It should also be clearly acknowledged when mentioning environment details that these are simple environments and are used as a proof of concept.
> >
> > [1] Hazan, Elad, et al. "Provably efficient maximum entropy exploration." International Conference on Machine Learning. PMLR, 2019.
> >
> > [2] Eysenbach, Benjamin, and Sergey Levine. "Maximum entropy RL (provably) solves some robust RL problems." arXiv preprint arXiv:2103.06257 (2021).
> >
> > [3] Eysenbach, Benjamin, et al. "Diversity is all you need: Learning skills without a reward function." arXiv preprint arXiv:1802.06070 (2018).

---

> > > ### Author Response · Authors · 2025-04-05
> > >
> > > Dear Reviewer,
> > >
> > > We are happy to see that our replies helped clarify important aspects of the paper. We thank the reviewer for raising the points in the first place and for giving us important feedback on what is not clear in the current writing. **We will incorporate all of these clarifications in an updated and improved version of the manuscript**.
> > >
> > > Regarding the **experiments**, it was not our intention to overstate their scope. This is why we named the experimental section "Empirical Corroboration" where "we report numerical experiments in simple yet illustrative domains" (lines 247-249).
> > > We acknowledge that the introduction is not as clear on the nature of the experiments.
> > > **Following the reviewer's suggestion**, we will change "We provide numerical experiments to corroborate our findings[...]" into "We provide numerical experiments in illustrative domains for a preliminary validation of our findings[...]" at line 107 of the introduction.
> > >
> > > We agree that a thorough empirical evaluation in **more challenging domains** would make for an even stronger submission. However, we note that **it is not standard in previous publications in the same area**. Works that are mainly conceptual and theoretical, e.g. Mutti et al 2020a, Guo et al 2021, Mutti et al 2022a, Tiapkin et al 2023, Zamboni et al 2024ab, only reports experiments in gridworld or chain MDPs, which are comparable with the domains we considered. Hazan et al 2019 is actually an exception, although their Mujoco experiments have been shown to lead to extremely sub-optimal results (see Mutti et al 2021, Liu and Abbeel 2021 for comparison with ``practical´´ algorithms). Regarding the other mentioned papers: [2] is tackling a different problem than ours (entropy of the policy, not the state visitation), whereas [3] is somewhat related (see our rebuttal to R. Fend above) but mostly empirical.
> > >
> > > Finally, we hope the additional clarifications have convinced the reviewer on the value of our work and to increase their score accordingly.
> > >
> > > [Mutti et al. An intrinsically-motivated approach for learning highly exploring and fast mixing
> > > policies, 2020]
> > >
> > > [Guo et al. Geometric entropic exploration, 2021]
> > >
> > > [Mutti et al. The importance of non-Markovianity in maximum state entropy exploration, 2022]
> > >
> > > [Tiapkin et al. Fast rates for maximum entropy exploration, 2023]
> > >
> > > [Zamboni et al. How to explore with belief: State entropy maximization in
> > > pomdps, 2024a]
> > >
> > > [Zamboni et al., The
> > > limits of pure exploration in pomdps: When the observation entropy is enough, 2024b]
> > >
> > > [Mutti et al., Task-agnostic
> > > exploration via policy gradient of a non-parametric state
> > > entropy estimate, 2021]
> > >
> > > [Liu and Abbeel, Behavior from the void: Unsupervised active pre-training, 2021]

---

### Official Review · Reviewer_okjP · 2025-03-14

**Overall Recommendation:** 4

**Summary:**

This paper studies how parallel training facilitates exploration in reinforcement learning. The major result is that parallel exploration can not only obtain batched acceleration compared to single-agent ones, but it’s also possible to further improve sample complexity though diversity-driven policy design. The results come from a novel and careful analysis of parallel exploration procedure and are backed up through sufficient empirical experiments.

**Claims And Evidence:**

Yes

**Essential References Not Discussed:**

Not I aware of.

**Experimental Designs Or Analyses:**

Yes

**Methods And Evaluation Criteria:**

Yes

**Other Comments Or Suggestions:**

1. For clarification, in line 204, the author claims: As a result, they induce distributions with  ‘lower’ entropy compared to a single policy covering the entire space. Can the authors elaborate on how to get this conclusion from Theorem 4.1?

2. Line 252, typo: double "the"

**Other Strengths And Weaknesses:**

### Strengths

This paper proposed a novel problem: if we can run n agents in parallel, how should we maximize the exploration (i.e., state occupancy entropy)?

Turns out the answer is not as trivial as running the same "uniform" policy for n agents simultaneously. A better policy is to run $n$ different policies that each maximize the exploration for a sub-region, and in aggregation, we achieve a uniform distribution overall. The reason is that running a sub-region can give us a better sample complexity in terms of estimating state transitions and probabilities. After seeing the explanation, it feels obvious but the mathematical derivations are carried out satisfyingly, in particular the decomposition insights.

What is more surprising is the policy gradient method proposed by the authors. Turns out that the gradient of the entropy objective for each single agent can be derived in a distributed manner (with aggregated empirical distribution). Therefore a practical algorithm can be carried in terms of maximizing exploration in a non-trivial way.

The paper is well-written. The model is clear and the analysis is intuitive, and experiments are conducted satisfyingly.

### Weakness

One question is that if the motivation is for the speed, how gradient descent step slow down the computation? This is not discussed in the paper.

**Questions For Authors:**

if the motivation is for the speed, how gradient descent step slow down the computation? Is that possible that we can synchronize the gradients once in a while so that parallelization can be maximized?

**Relation To Broader Scientific Literature:**

This result seems to be of broader interest in RL as more and more vectorized/parallel environment simulation is available in recent literature.

**Theoretical Claims:**

Yes

---

> ### Author Rebuttal · Authors · 2025-03-31
>
> We sincerely appreciate the reviewer's time and effort in evaluating our work and are grateful for the opportunity to provide further clarifications. We also thank the reviewer for pointing out the typo errors; we will address them in the final version.
>
> **If the motivation is for the speed, how gradient descent step slow down the computation?**
>
> We are not fully sure to understand reviewer's question here. We provide a tentative reply below. If the reviewer feel we are not addressing their point, we will be more than happy to provide further considerations.
>
> The discussion on the cost of the gradient calculation is an excellent point. We have not analyzed the computational cost of calculating the gradient at each step, which is negligible in our experiments. However, in more complex domains, a vectorized calculation of the gradient (which is what the reviewer suggests?) will definitely be beneficial from the computational cost perspective. We will provide this important consideration in the manuscript.
>
> **As a result, they induce distributions with ‘lower’ entropy compared to a single policy covering the entire space. Can the authors elaborate on how to get this conclusion from Theorem 4.1?**
>
> We acknowledge that we may not have fully clarified the use of the findings from Theorem 4.1, thus we will provide extended intuition in a revised version of the manuscript.
>
> We designed Equation 1 to intrinsically motivate each agent to reduce the portion of the state space it explores, encouraging diversity. This leads to an induced empirical state distribution with lower entropy due to the small size of the support of the states visited by each single agent.
>
> As outlined in Theorem 4.1, a lower entropy state distribution means that fewer samples are needed to realize the target distribution approximately.

---

### Official Review · Reviewer_58WA · 2025-03-14

**Overall Recommendation:** 3

**Summary:**

This paper proposes an exploration framework for parallel agents with state entropy maximization and an analysis of the framework. They showed on tabular environments that parallel exploration covers the state space better than single-agent exploration with the same compute budget. And datasets collected by parallel agents also help post offline training.

**Claims And Evidence:**

Yes.

**Essential References Not Discussed:**

Entropy-based diversity is not something new for multi-agent RL. See [1]  Lupu, Andrei, et al. "Trajectory diversity for zero-shot coordination." [2] Zhao, Rui, et al. "Maximum entropy population-based training for zero-shot human-ai coordination."

**Experimental Designs Or Analyses:**

I reviewed the experiments and they appear sound and well-reasoned.

**Methods And Evaluation Criteria:**

Yes.

**Other Comments Or Suggestions:**

No.

**Other Strengths And Weaknesses:**

The paper is well-written. The experiments provide strong support for the hypothesis that parallel exploration is more efficient. While this work focuses on tabular environments, it would be interesting to see if the algorithm and analysis can be extended to a larger state space possibly with function approximation.
The hypothesis that parallel exploration is more efficient because each agent can focus on a smaller region is interesting and intuitive. This is also insightful for future works on exploration. I believe it is true, but I would like to see more support for the hypothesis.

**Questions For Authors:**

The ``Main Takeout’’ section argues that each agent can focus on different regions. Why are the agents incentivized to be different rather than all become the same policy with uniform coverage over states? Is it because if the entropy of each agent’s distribution is low, then the empirical estimation error will be lower? Is the efficiency of parallel exploration better than single-agent exploration shown by sample efficiency or convergence rate?
2. In parallel exploration, how does the entropy of each individual dataset look like and how different (e.g., KL divergence) are the datasets between different agents?
3. In line 240-247, why can the log tricks be applied when $d_p$ is the mixture distribution, which does not only depend on $\pi_i$?

**Relation To Broader Scientific Literature:**

This paper provides proof-of-concept analysis for parallel exploration, which is relatively new and important to the key exploration problem for RL.

**Theoretical Claims:**

The derivation In lines 240-247 needs further explanation.
I did not check Theorem A.2.

---

> ### Author Rebuttal · Authors · 2025-03-31
>
> We sincerely appreciate the reviewer's time and effort in evaluating our work, and we are grateful for the opportunity to provide further clarifications.
>
> **Why are the agents incentivized to be different rather than all become the same policy with uniform coverage over states? Is it because if the entropy of each agent’s distribution is low, then the empirical estimation error will be lower? Is the efficiency of parallel exploration better than single-agent exploration shown by sample efficiency or convergence rate?**
>
> Precisely, the reviewer is right: If each agent's policy has low entropy, then the empirical realization will concentrate faster to the target distribution. This finding is supported by Theorem 4.1. Since the entropy term appears in the numerator of the samples lower bound, we will need fewer samples (smaller *n*) to concentrate around the target distribution. Even if a policy with uniform expected coverage over the states exists (it does not typically), a single realization from this policy may not have uniform coverage. Appropriately specialized policy can still be preferable.
>
> Consider a two-room gridworld with two specialized parallel agents, each assigned to a specific room. In a single realization, the state distribution will be uniform because each agent deterministically explores its designated room.
> In contrast, a single agent following a maximum entropy policy has a higher variance in its state distribution. This means that it is not guaranteed to visit both rooms in a single realization, as its exploration is probabilistic rather than deterministic.
>
> **In parallel exploration, how does the entropy of each individual dataset look like and how different (e.g., KL divergence) are the datasets between different agents?**
>
> Thanks to the insightful question posed by the reviewer, we can further clarify that under this objective formulation, each learner is intrinsically encouraged to minimize entropy along its own trajectory. This naturally leads to the emergence of specialized policies across different regions of the state space, as illustrated in Figure 10. Indeed in the same figure where the action distributions are plotted, in the parallel case each agent tends to follow a more deterministic policy, naturally dividing the state space among the *m* agents. In contrast, a single agent aiming to maximize the entropy of its state distribution spreads its actions more uniformly.
>
> **In line 240-247, why can the log tricks be applied when $d_p$ is the mixture distribution, which does not only depend on $π_i$?**
>
> Since agents have independent policies, the gradient of each agent’s policy with respect to its parameters depends only on the states it has visited itself. Specifically, changing the policy parameter $θ_i$ of the *i*-th agent does not influence the trajectory generated by others.
>
> **Missing Reference**
>
> Regarding the references, we thank the reviewer for pointing out the interesting papers about coordination, which we will mention in an updated version of the manuscript. Below we explain how they differ from our work:
>
> - **Trajectory Diversity for Zero-Shot Coordination:** We note that the paper addresses the problem of zero-shot coordination in a setting that is related to MARL. In MARL, multiple agents are acting in the *same* environment, where state transitions depends on the joint action $a_t = (a_{1t}, ..., a_{kt})$. In contrast, in our setting, multiple agents interact with *independent* copies of the environments, where the trajectories experienced are also independent. Moreover, even if in the *Diversity* objective considered in Section 4.2 and ours have some similarity in nature, their objective is not intended to maximize entropy, but only diversity among the policies. We appreciate the reviewer’s suggestion and will cite the paper in the final version to clarify our position.
>
> - **Maximum Entropy Population-Based Training for Zero-Shot Human-AI Coordination:** As in the previous example, the main difference with this work is related to the construction of the environment. The definition of the *Two-Player MDP* positions this paper in the MARL setting, where the agents create dependent trajectories. For completeness in positioning, our paper will include it within our reference list.

---

### Official Review · Reviewer_Fend · 2025-03-15

**Overall Recommendation:** 3

**Summary:**

This paper investigates how to effectively maximize state entropy exploration in parallel agent settings. The authors propose a framework where multiple agents, each operating in separate environment replicas, are trained to collectively maximize the entropy of their visited state distribution while promoting diversity among their exploration strategies. They introduce a parallel learning objective that explicitly balances individual agent entropy with inter-agent diversity, and develop a policy gradient algorithm (PGPSE) to optimize this objective. The paper includes theoretical analysis on concentration properties showing that parallel agents with diverse policies can achieve faster convergence to high-entropy distributions compared to single agents. Experimental results on gridworld environments demonstrate that the proposed approach outperforms single-agent baselines in terms of state entropy, support size, and performance in downstream offline RL tasks.

**Claims And Evidence:**

The paper's claims about the benefits of parallel exploration with diverse agents are generally supported by the theoretical and empirical evidence presented. However, the claim about the superiority of their approach over existing methods is not fully supported due to limited comparison with relevant baselines. While they show improvement over single-agent exploration and random policies, they don't compare against other methods specifically designed to promote diversity among agents (like DIAYN or diversity-promoting MARL approaches). This is a limitation in evaluating the novelty and effectiveness of their contribution.

**Essential References Not Discussed:**

It is quite appropriate, and it will be good to discuss Celebrating Diversity in Shared Multi-Agent Reinforcement Learning.

**Experimental Designs Or Analyses:**

The experimental design is generally sound but limited in scope. However, the experiments are confined to relatively simple gridworld environments, and the paper lacks comparison with more relevant baselines that specifically address diversity among agents. This makes it difficult to assess the true novelty and contribution of the proposed approach relative to existing methods.

**Methods And Evaluation Criteria:**

The proposed methods for parallel state entropy maximization make sense for the problem at hand. The evaluation criteria focusing on normalized entropy, support size, and downstream offline RL performance are appropriate metrics for exploration quality. The environments used (gridworlds with different complexity levels) allow for clear visualization and interpretation of results.

However, the paper  only consider simple discrete gridworld tasks and did not consider more challenging or continuous domains. The evaluation would be more convincing if it included more challenging environments beyond gridworlds and compared against state-of-the-art exploration methods that explicitly promote diversity, such as DIAYN or diversity-centered MARL approaches like "Celebrating Diversity in Shared Multi-Agent Reinforcement Learning."

**Other Comments Or Suggestions:**

- Consider expanding the experimental evaluation to include more complex environments.
- The visualization in Figure 10 showing the different policies learned by parallel agents is interesting, but could benefit from more quantitative analysis of the diversity.

**Other Strengths And Weaknesses:**

Strengths:
- The theoretical analysis of concentration properties provides useful insights into why parallel diversity is beneficial.
- The formulation of the parallel learning objective is elegant and intuitive.
- The visualization of learned policies and datasets helps in understanding the behavior of the method.

Weaknesses:
- Limited comparison with relevant baselines that specifically address diversity among agents.
- Experiments are confined to relatively simple gridworld environments.

**Questions For Authors:**

- Why didn't the authors compare the proposed approach with diversity-promoting methods like DIAYN or diversity-centered MARL approaches? Such comparisons would provide stronger evidence for the novelty and effectiveness of the approach.
- How would the method perform in more complex environments beyond gridworlds? The current environments are too simple to fully demonstrate the benefits of parallel diversity.
- Have the authors considered extending your approach to continuous state and action spaces, which are common in real-world applications like robotics?

**Relation To Broader Scientific Literature:**

The paper builds upon prior work on state entropy maximization for exploration in reinforcement learning, particularly in the reward-free setting. It extends this concept to parallel settings, which is a relevant direction given the increasing use of parallelization in modern RL systems.

However, the paper doesn't adequately position itself relative to literature on diversity-promoting exploration strategies. Particularly missing is comparison with methods like DIAYN which explicitly maximizes diversity among skills/policies, or diversity-centered MARL approaches like "Celebrating Diversity in Shared Multi-Agent Reinforcement Learning" which explicitly aims to increase diversity among agents' behaviors.

**Theoretical Claims:**

The theoretical analysis appears sound.

---

> ### Author Rebuttal · Authors · 2025-03-31
>
> We sincerely appreciate the reviewer's time and effort in evaluating our work and greatly value their insightful comments and suggestions. To clarify the key points of discussion and our design choices, we provide the following responses.
>
> **Why didn’t the authors compare the proposed approach with diversity-promoting methods like DIAYN or diversity-centered MARL approaches?**
>
> - **MARL:** We want to underline an important difference between the typical MARL setting and ours. In MARL, multiple agents are acting in the *same* environment. They need coordination because their trajectories are dependent. In our setting, multiple agents interact with *independent* copies of the environments. We show that coordination is useful in specializing the objective of each single agent, but their trajectories are independent. This makes the two settings significantly different.
>
> - **Celebrating Diversity in Shared Multi-Agent Reinforcement Learning:** While the paper introduces policy diversification via mutual information maximization, it remains within the MARL framework, which we aim to differentiate from. Since the agents share policy parameters in a non-reward-free setting, we did not initially consider it. The key distinction is that our agents are fully independent, driven solely by the entropy of the mixture state distribution. We appreciate the reviewer’s suggestion and will cite the paper in the final version to clarify our position.
>
> - **DIAYN:** We note that the intended use of DIAYN is substantially different than our setting. In the original paper, the authors use DIAYN to learn diverse *skills*, which are specialized policies or options that may be combined in the same environment to achieve complex goals. Here, we want to learn diverse policies to collect maximum entropy data across parallel simulators. However, we agree with the reviewer that DIAYN could be adapted to our setting, making the comparison potentially interesting. We will include a comparison with DIAYN in a revised version of the manuscript.
>
> **How would the method perform in more complex environments beyond gridworlds? Have the authors considered extending the approach to continuous state and action spaces?**
>
> We fully agree on the importance of extending our approach to continuous state and action spaces, which we see as a natural direction for future work. Our PGPSE algorithm can be adapted to these settings by computing trajectory entropy using non-parametric estimators, as explored in works like *Task-Agnostic Exploration via Policy Gradient of a Non-Parametric State Entropy Estimate* and *Behavior From the Void: Unsupervised Active Pre-Training*. This work serves as a foundational step in advancing parallel learning, focusing on defining the setting and establishing the theoretical basis for replica learners.

---

### Decision · Program_Chairs · 2025-05-01

**Decision:**

Accept (poster)

**Comment:**

This paper studies generating diverse experience for policy gradient algorithms in reward-free settings through the use of entropy maximisation and parallel RL training to facilitates exploration.

Reviewers okjP, Fend and 58WA praised the problem formulation of maximising the exploration of multiple agents in parallel over subregions, and the mathematical derivation. Reviewer okjP praised the writing and clarity of the paper. Reviewer khWs found the application to offline RL interesting.

Reviewer khWs had multiple concerns about the empirical evidence provided in the paper, including on the simplicity of the environments and lack of baseline, but these were addressed during rebuttal. Reviewer 58WA suggested using larger state spaces for exploration. Reviewer Fend and the authors engaged in a discussion about the relevance of a few MARL-based references.

Based on the average score of 3.25, I believe that the paper should be accepted.